# A Framework for Fast and Stable Representations of Multiparameter Persistent Homology Decompositions

**David Loiseaux**
DataShape
Centre Inria d'Université Côte d'Azur
Biot, France

**Mathieu Carrière**
DataShape
Centre Inria d'Université Côte d'Azur
Biot, France

**Andrew J. Blumberg**
Irving Institute for Cancer Dynamics
Columbia University
New-York, NY, USA

## Abstract

Topological data analysis (TDA) is an area of data science that focuses on using invariants from algebraic topology to provide multiscale shape descriptors for geometric data sets, such as graphs and point clouds. One of the most important such descriptors is *persistent homology*, which encodes the change in shape as a filtration parameter changes; a typical parameter is the feature scale. For many data sets, it is useful to simultaneously vary multiple filtration parameters, for example feature scale and density. While the theoretical properties of single parameter persistent homology are well understood, less is known about the multiparameter case. In particular, a central question is the problem of representing multiparameter persistent homology by elements of a vector space for integration with standard machine learning algorithms. Existing approaches to this problem either ignore most of the multiparameter information to reduce to the one-parameter case or are heuristic and potentially unstable in the face of noise. In this article, we introduce a new general representation framework that leverages recent results on *decompositions* of multiparameter persistent homology. This framework is rich in information, fast to compute, and encompasses previous approaches. Moreover, we establish theoretical stability guarantees under this framework as well as efficient algorithms for practical computation, making this framework an applicable and versatile tool for analyzing geometric data. We validate our stability results and algorithms with numerical experiments that demonstrate statistical convergence, prediction accuracy, and fast running times on several real data sets.

## 1 Introduction

Topological Data Analysis (TDA) [8] is a methodology for analyzing data sets using multiscale shape descriptors coming from algebraic topology. There has been intense interest in the field in the last decade, since topological features promise to allow practitioners to compute and encode information that classical approaches do not capture. Moreover, TDA rests on solid theoretical grounds, with guarantees accompanying many of its methods and descriptors. TDA has proved useful in a wide variety of application areas, including computer graphics [13, 33], computational biology [34], and material science [6, 35], among many others.

The main tool of TDA is *persistent homology*. In its most standard form, one is given a finite metric space $X$ (e.g., a finite set of points and their pairwise distances) and a continuous function

37th Conference on Neural Information Processing Systems (NeurIPS 2023).

$f : X \to \mathbb{R}$. This function usually represents a parameter of interest (such as, e.g., scale or density for point clouds, marker genes for single-cell data, etc), and the goal of persistent homology is to characterize the topological variations of this function on the data at all possible scales. Of course, the idea of considering multiscale representations of geometric data is not new [14, 32, 41]; the contribution of persistent homology is to obtain a novel and theoretically tractable multiscale shape descriptor. More formally, persistent homology is achieved by computing the so-called *persistence barcode* of $f$, which is obtained by looking at all sublevel sets of the form $\{f^{-1}((-\infty, \alpha])\}_{\alpha \in \mathbb{R}}$, also called *filtration induced by* $f$, and by computing a *decomposition* of this filtration, that is, by recording the appearances and disappearances of topological features (connected components, loops, enclosed spheres, etc) in these sets. When such a feature appears (resp. disappears), e.g., in a sublevel set $f^{-1}((-\infty, \alpha_b])$, we call the corresponding threshold $\alpha_b$ (resp. $\alpha_d$) the *birth time* (resp. *death time*) of the topological feature, and we summarize this information in a set of intervals, or bars, called the persistence barcode $D(f) := \{(\alpha_b, \alpha_d)\}_{\alpha \in A} \subset \mathbb{R} \times \mathbb{R} \cup \{\infty\}$. Moreover, the bar length $\alpha_d - \alpha_b$ often serves as a proxy for the statistical significance of the corresponding feature.

However, an inherent limitation of the formulation of persistent homology is that it can handle only a single filtration parameter $f$. However, in practice it is common that one has to deal with multiple parameters. This translates into multiple filtration functions: a standard example is when one aims at obtaining meaningful topological representation of a noisy point cloud. In this case, both feature scale and density functions are necessary (see Appendix A). An extension of persistent homology to several filtration functions is called *multiparameter* persistent homology [3, 9], and studies the topological variations of a continuous *multiparameter* function $f : X \to \mathbb{R}^n$ with $n \in \mathbb{N}^*$. This setting is notoriously difficult to analyze theoretically as there is no result ensuring the existence of an analogue of persistence barcodes, i.e., a decomposition into subsets of $\mathbb{R}^n$, each representing the lifespan of a topological feature.

Still, it remains possible to define weaker topological invariants in this setting. The most common one is the so-called *rank invariant* (as well as its variations, such as the generalized rank invariant [24], and its decompositions, such as the signed barcodes [4]), which describes how the topological features associated to any pair of sublevel sets $\{x \in X : f(x) \le \alpha\}$ and $\{x \in X : f(x) \le \beta\}$ such that $\alpha \le \beta$ (w.r.t. the partial order in $\mathbb{R}^n$), are connected. The rank invariant is a construction in abstract algebra, and so the task of finding appropriate *representations* of this invariant, i.e., embeddings into Hilbert spaces, is critical. Hence, a number of such representations have been defined, which first approximate the rank invariant by computing persistence barcodes from several linear combinations of filtrations, a procedure often referenced as the *fibered barcode* (see Appendix E), and then aggregate known single-parameter representations for them [17, 18, 39]. Adequate representations of the generalized rank invariant have also been investigated recently for $n = 2$ [42].

However, the rank invariant, and its associated representations, are known to be much less informative than decompositions (when they exist): many functions have different decompositions yet the same rank invariants. Therefore, the aforementioned representations can encode only limited multiparameter topological information. Instead, in this work, we focus on *candidate decompositions* of the function, in order to create descriptors that are strictly more powerful than the rank invariant. Indeed, while there is no general decomposition theorem, there is recent work that constructs candidate decompositions in terms of simple pieces [1, 7, 29] that always exist but do not necessarily suffice to reconstruct all of the multiparameter information. Nonetheless, they are strictly more informative than the rank invariant under mild conditions, are stable, and approximate the true decomposition when it exists[1]. For instance, in Figure 2, we present a bifiltration of a noisy point cloud with scale and density (**left**), and a corresponding candidate decomposition comprised of subsets of $\mathbb{R}^2$, each representing a topological feature (**middle**). One can see that there is a large green subset in the decomposition that represents the circle formed by the points that are not outliers (also highlighted in green in the bifiltration).

---

[1]Although multiparameter persistent homology can always be decomposed as a sum of indecomposable pieces ([3, Theorem 4.2], [20]), these decompositions are prohibitively difficult to interpret and work with.

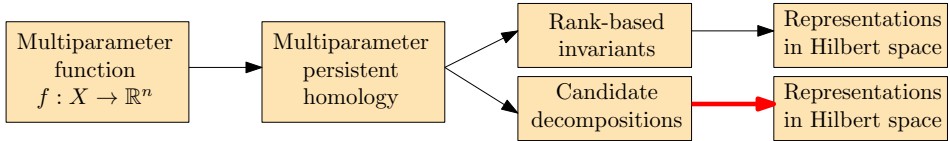

Figure 1: Common pipelines for the use of multiparameter persistent homology in data science—our work provides new contributions to the arrow highlighted in red.

Unfortunately, while more informative, candidate decompositions suffer from the same problem than the rank invariant; they also need appropriate representations in order to be processed by standard data science methods. In this work, we bridge this gap by providing new representations designed for candidate decompositions. See Figure 1 for a summarizing figure.

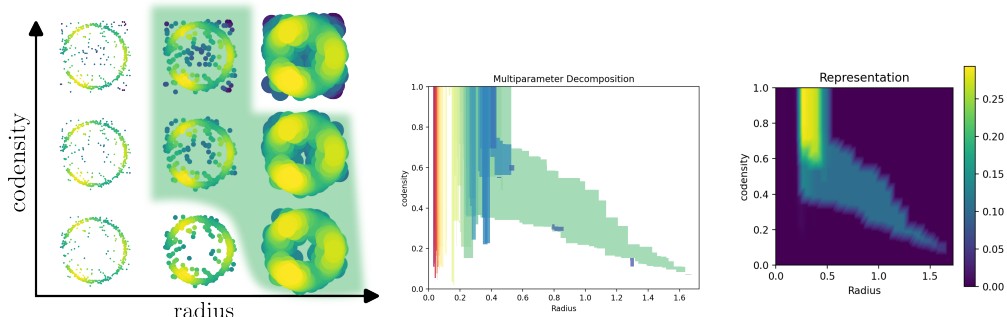

Figure 2: **(left)** Bi-filtration of a noisy point cloud induced by both feature scale (using unions of balls with increasing radii) and sublevel sets of codensity. The cycle highlighted in the green zone can be detected as a large subset in the corresponding candidate decomposition computed by the MMA method [29] **(middle)**, and in our representation of it **(right)**.

**Contributions.** Our contributions in this work are listed below:

- We provide a general framework that parametrizes representations of multiparameter persistent homology decompositions (Definition 1) and which encompasses previous approaches in the literature. These representations take the form of a parametrized family of continuous functions on $\mathbb{R}^n$ that can be binned into images for visualization and data science.

- We identify parameters in this framework that result in representations that have stability guarantees while still encoding more information than the rank invariant (see Theorem 1).

- We illustrate the performance of our framework with numerical experiments: (1) We demonstrate the practical consequences of the stability theorem by measuring the statistical convergence of our representations. (2) We achieve the best performance with the lowest runtime on several classification tasks on public data sets (see Sections 4.1 and 4.2).

**Related work.** Closely related to our method is the recent contribution [10], which also proposes a representation for decompositions. However, their approach, while being efficient in practice, is a heuristic with no corresponding mathematical guarantees. In particular, it is known to be unstable: similar decompositions can lead to very different representations, as shown in Appendix B. Our approach can be understood as a subsequent generalization of the work of [10], with new mathematical guarantees that allow to derive, e.g., statistical rates of convergence.

**Outline.** Our work is organized as follows. In Section 2, we recall the basics of multiparameter persistent homology. Next, in Section 3 we present our general framework and state our associated stability result. Finally, we showcase the numerical performances of our representations in Section 4, and we conclude in Section 5.

## 2 Background

In this section, we briefly recall the basics of single and multiparameter persistent homology, and refer the reader to Appendix C, Appendix D, and [31, 34] for a more complete treatment.

**Persistent homology.** The basic brick of persistent homology is a *filtered topological space $X$*, by which we mean a topological space $X$ together with a function $f: X \to \mathbb{R}$ (for instance, in Figure 5, $X = \mathbb{R}^2$ and $f = f_P$). Then, given $\alpha > 0$, we call $F(\alpha) := f^{-1}((-\infty, \alpha]) \subseteq X$ the *sublevel set of $f$ at level $\alpha$*. Given levels $\alpha_1 \leq \cdots \leq \alpha_N$, the corresponding sublevel sets are nested w.r.t. inclusion, i.e., one has $F(\alpha_1) \subseteq F(\alpha_2) \subseteq \ldots \subseteq F(\alpha_i) \subseteq \ldots \subseteq F(\alpha_N)$. This system is an example of *filtration* of $X$, where a filtration is generally defined as a sequence of nested subspaces $X_1 \subseteq \ldots \subseteq X_i \subseteq \ldots \subseteq X$. Then, the core idea of persistent homology is to apply the $k$th *homology functor $H_k$* on each $F(\alpha_i)$. We do not define the homology functor explicitly here, but simply recall that each $H_k(F(\alpha_i))$ is a vector space, whose basis elements represent the $k$th dimensional topological features of $F(\alpha_i)$ (connected components for $k = 0$, loops for $k = 1$, spheres for $k = 2$, etc). Moreover, the inclusions $F(\alpha_i) \subseteq F(\alpha_{i+1})$ translate into linear maps $H_k(F(\alpha_i)) \to H_k(F(\alpha_{i+1}))$, which connect the features of $F(\alpha_i)$ and $F(\alpha_{i+1})$ together. This allows to keep track of the topological features in the filtration, and record their levels, often called times, of appearance and disappearance. More formally, such a sequence of vector spaces connected with linear maps $\mathbb{M} = H_*(F(\alpha_1)) \to \cdots \to H_*(F(\alpha_N))$ is called a *persistence module*, and the standard decomposition theorem [15, Theorem 2.8] states that this module can always be decomposed as $\mathbb{M} = \oplus_{i=1}^m \mathbb{I}[\alpha_{b_i}, \alpha_{d_i}]$, where $\mathbb{I}[\alpha_{b_i}, \alpha_{d_i}]$ stands for a module of dimension 1 (i.e., that represents a single topological feature) between $\alpha_{b_i}$ and $\alpha_{d_i}$, and dimension 0 (i.e., that represents no feature) elsewhere. It is thus convenient to summarize such a module with its *persistence barcode* $D(\mathbb{M}) = \{[\alpha_{b_i}, \alpha_{d_i}]\}_{1 \leq i \leq m}$. Note that in practice, one is only given a sampling of the topological space $X$, which is usually unknown. In that case, persistence barcodes are computed using combinatorial models of $X$ computed from the data, called *simplicial complexes*. See Appendix C.

**Multiparameter persistent homology.** The persistence modules defined above extend straightforwardly when there are multiple filtration functions. An $n$-filtration, or multifiltration, induced by a function $f : X \to \mathbb{R}^n$, is the family of sublevel sets $F = \{F(\alpha)\}_{\alpha \in \mathbb{R}^n}$, where $F(\alpha) := \{x \in X : f(x) \leq \alpha\}$ and $\leq$ denotes the partial order of $\mathbb{R}^n$. Again, applying the homology functor $H_k$ on the multifiltration $F$ induces a *multiparameter persistence module* $\mathbb{M}$. However, contrary to the single-parameter case, the algebraic structure of such a module is very intricate, and there is no general decomposition into modules of dimension at most 1, and thus no analogue of the persistence barcode. Instead, the *rank invariant* has been introduced as a weaker invariant: it is defined, for a module $\mathbb{M}$, as the function $\mathrm{RI} : (\alpha, \beta) \mapsto \mathrm{rank}(\mathbb{M}(\alpha) \to \mathbb{M}(\beta))$ for any $\alpha \leq \beta$, but is also known to miss a lot of structural properties of $\mathbb{M}$. To remedy this, several methods have been developed to compute *candidate decompositions* for $\mathbb{M}$ [1, 7, 29], where a candidate decomposition is a module $\tilde{\mathbb{M}}$ that can be decomposed as $\tilde{\mathbb{M}} \simeq \oplus_{i=1}^m M_i$, where each $M_i$ is an *interval module*, i.e., its dimension is at most 1, and its support $\mathrm{supp}(M_i) := \{\alpha \in \mathbb{R}^n : \dim(M_i(\alpha)) = 1\}$ is an interval of $\mathbb{R}^n$ (see Appendix D). In particular, when $\mathbb{M}$ does decompose into intervals, candidate decompositions must agree with the true decomposition. One also often asks candidate decompositions to preserve the rank invariant.

**Distances.** Finally, multiparameter persistence modules can be compared with two standard distances: the *interleaving* and *bottleneck* (or $\ell^\infty$) distances. Their explicit definitions are technical and not necessary for our main exposition, so we refer the reader to, e.g., [3, Sections 6.1, 6.4] and Appendix D for more details. The *stability theorem* [27, Theorem 5.3] states that multiparameter persistence modules are stable: $d_I(\mathbb{M}, \mathbb{M}') \leq \|f - f'\|_\infty$, where $f$ and $f'$ are continuous multiparameter functions associated to $\mathbb{M}$ and $\mathbb{M}'$ respectively.

## 3 T-CDR: a template for representations of candidate decompositions

Even though candidate decompositions of multiparameter persistence modules are known to encode useful data information, their algebraic definitions make them not suitable for subsequent data science and machine learning purposes. Hence, in this section, we introduce the Template Candidate

Decomposition Representation (T-CDR): a general framework and template system for representations of candidate decompositions, i.e., maps defined on the space of candidate decompositions and taking values in an (implicit or explicit) Hilbert space.

## 3.1 T-CDR definition

**Notations.** In this article, by a slight abuse of notation, we will make no difference in the notations between an interval module and its support, and we will denote the restriction of an interval support $M$ to a given line $\ell$ as $M\big|_\ell$.

**Definition 1.** Let $\mathbb{M} = \oplus_{i=1}^m M_i$ be a candidate decomposition, and let $\mathcal{M}$ be the space of interval modules. The *Template Candidate Decomposition Representation* (T-CDR) of $\mathbb{M}$ is:

$$V_{\mathrm{op},w,\phi}(\mathbb{M}) = \mathrm{op}(\{w(M_i) \cdot \phi(M_i)\}_{i=1}^m), \tag{1}$$

where $\mathrm{op}$ is a permutation invariant operation (sum, max, min, mean, etc), $w : \mathcal{M} \to \mathbb{R}$ is a weight function, and $\phi : \mathcal{M} \to \mathcal{H}$ sends any interval module to a vector in a Hilbert space $\mathcal{H}$.

The general definition of T-CDR is inspired from a similar framework that was introduced for single-parameter persistence with the automatic representation method *PersLay* [11].

**Relation to previous work.** Interestingly, whenever applied on candidate decompositions that preserve the rank invariant, specific choices of $\mathrm{op}$, $w$ and $\phi$ reproduce previous representations:

- Using $w : M_i \mapsto 1$, $\phi : M_i \mapsto \begin{cases} \mathbb{R}^n & \to \mathbb{R} \\ x & \mapsto \Lambda(x, M_i\big|_{\ell_x}) \end{cases}$ and $\mathrm{op} = k$th maximum, where $l_x$ is the diagonal line crossing $x$, and $\Lambda(\cdot, \ell)$ denotes the tent function associated to any segment $\ell \subset \mathbb{R}^n$, induces the $k$th multiparameter persistence landscape (MPL) [39].

- Using $w : M_i \mapsto 1$, $\phi : M_i \mapsto \begin{cases} \mathbb{R}^n \times \mathbb{R}^n & \to \mathbb{R}^d \\ p,q & \mapsto w'(M_i \cap [p,q]) \cdot \phi'(M_i \cap [p,q]) \end{cases}$ and $\mathrm{op} = \mathrm{op}'$, where $\mathrm{op}'$, $w'$ and $\phi'$ are the parameters of any persistence diagram representation from Perslay, induces the multiparameter persistence kernel (MPK) [17].

- Using $w : M_i \mapsto \mathrm{vol}(M_i)$, $\phi : M_i \mapsto \begin{cases} \mathbb{R}^n & \to \mathbb{R} \\ x & \mapsto \exp(-\min_{\ell \in L} d(x, M_i\big|_\ell)^2/\sigma^2) \end{cases}$ and $\mathrm{op} = \sum$, where $L$ is a set of (pre-defined) diagonal lines, induces the multiparameter persistence image (MPI) [10].

Recall that the first two approaches are built from fibered barcodes and rank invariants, and that it is easy to find persistence modules that are different yet share the same rank invariant (see [38, Figure 3]). On the other hand, the third approach uses more information about the candidate decomposition, but is known to be unstable (see Appendix B). Hence, in the next section, we focus on specific choices for the T-CDR parameters that induce stable yet informative representations.

## 3.2 Metric properties

In this section, we study specific parameters for T-CDR (see Definition 1) that induce representations with associated robustness properties. We call this subset of representations *Stable Candidate Decomposition Representations* (S-CDR), and define them below.

**Definition 2.** The S-CDR parameters are:

1. the weight function $w : M \mapsto \sup\{\varepsilon > 0 : \exists y \in \mathbb{R}^n \text{ s.t. } \ell_{y,\varepsilon} \subset \mathrm{supp}\,(M)\}$, where $\ell_{y,\varepsilon}$ is the segment between $y - \varepsilon \cdot [1,\dots,1]$ and $y + \varepsilon \cdot [1,\dots,1]$,

2. the individual interval representations $\phi_\delta(M) : \mathbb{R}^n \to \mathbb{R}$:

    (a) $\phi_\delta(M)(x) = \frac{1}{\delta} w(\mathrm{supp}\,(M) \cap R_{x,\boldsymbol\delta})$,   (b) $\phi_\delta(M)(x) = \frac{1}{(2\delta)^n} \mathrm{vol}\,(\mathrm{supp}\,(M) \cap R_{x,\boldsymbol\delta})$,

    (c) $\phi_\delta(M)(x) = \frac{1}{(2\delta)^n} \sup_{x',\boldsymbol\delta'} \{\mathrm{vol}(R_{x',\boldsymbol\delta'}) : R_{x',\boldsymbol\delta'} \subseteq \mathrm{supp}\,(M) \cap R_{x,\boldsymbol\delta}\}$,

    where $R_{x,\boldsymbol\delta}$ is the hypersquare $\{y \in \mathbb{R}^n : x - \boldsymbol\delta \leq y \leq x + \boldsymbol\delta\} \subseteq \mathbb{R}^n$, $\boldsymbol\delta := \delta \cdot [1,\dots,1] \in \mathbb{R}^n$ for any $\delta > 0$, and $\mathrm{vol}$ denotes the volume of a set in $\mathbb{R}^n$.

3. the permutation invariant operators $\mathrm{op} = \sum$ and $\mathrm{op} = \sup$.

Intuitively, the S-CDR weight function is the length of the largest diagonal segment one can fit inside $\mathrm{supp}(M)$, and the S-CDR interval representations (a), (b) and (c) are the largest normalized diagonal length, volume, and hypersquare volume that one can fit inside $\mathrm{supp}(M) \cap R_{x,\boldsymbol{\delta}}$, respectively. These S-CDR interval representations allow for some trade-off between computational cost and the amount of information that is kept: (a) and (c) are very easy to compute, but (b) encodes more information about interval shapes. See Figure 2 (right) for visualizations.

Equipped with these S-CDR parameters, we can now define the two following S-CDRs, that can be applied on any candidate decomposition $\mathbb{M} = \oplus_{i=1}^m M_i$:

$$V_{p,\delta}(\mathbb{M}) := \sum_{i=1}^m \frac{w(M_i)^p}{\sum_{j=1}^m w(M_j)^p} \phi_\delta(M_i), \quad (2) \qquad\qquad V_{\infty,\delta}(\mathbb{M}) := \sup_{1 \leq i \leq m} \phi_\delta(M_i). \quad (3)$$

**Stability.**    The main motivation for introducing S-CDR parameters is that the corresponding S-CDRs are stable in the interleaving and bottleneck distances, as stated in the following theorem.

**Theorem 1.** *Let $\mathbb{M} = \oplus_{i=1}^m M_i$ and $\mathbb{M}' = \oplus_{j=1}^{m'} M_j'$ be two candidate decompositions. Assume that we have $\frac{1}{m} \sum_i w(M_i), \frac{1}{m'} \sum_j w(M_j') \geq C$, for some $C > 0$. Then for any $\delta > 0$, one has*

$$\|V_{0,\delta}(\mathbb{M}) - V_{0,\delta}(\mathbb{M}')\|_\infty \leq 2(d_{\mathrm{B}}(\mathbb{M}, \mathbb{M}') \wedge \delta)/\delta, \quad (4)$$

$$\|V_{1,\delta}(\mathbb{M}) - V_{1,\delta}(\mathbb{M}')\|_\infty \leq \left[4 + \frac{2}{C}\right](d_{\mathrm{B}}(\mathbb{M}, \mathbb{M}') \wedge \delta)/\delta, \quad (5)$$

$$\|V_{\infty,\delta}(\mathbb{M}) - V_{\infty,\delta}(\mathbb{M}')\|_\infty \leq (d_{\mathrm{I}}(\mathbb{M}, \mathbb{M}') \wedge \delta)/\delta, \quad (6)$$

*where $\wedge$ stands for minimum.*

A proof of Theorem 1 can be found in Appendix F.

These results are the main theoretical contribution in this work, as the only other decomposition-based representation in the literature [10] has no such guarantees. The other representations [17, 18, 39, 42] enjoy similar guarantees than ours, but are computed from the rank invariant and do not exploit the information contained in decompositions. Theorem 1 shows that S-CDRs bring the best of both worlds: these representations are richer than the rank invariant and stable at the same time. We also provide an additional stability result with a similar, yet more complicated representation in Appendix G, whose upper bound does not involve taking minimum.

**Remark 1.** S-CDRs are injective representations: if the support of two interval modules are different, then their corresponding S-CDRs (evaluated on a point that belongs to the support of one interval but not on the support of the other) will differ, provided that $\delta$ is sufficiently small.

## 4    Numerical Experiments

In this section, we illustrate the efficiency of our S-CDRs with numerical experiments. First, we explore the stability theorem in Section 4.1 by studying the convergence rates, both theoretically and empirically, of S-CDRs on various data sets. Then, we showcase the efficiency of S-CDRs on classification tasks in Section 4.2, and we investigate their running times in Section 4.3. Our code for computing S-CDRs is based on the `MMA` [29] and `Gudhi` [36] libraries for computing candidate decompositions[2]. It is publicly available at https://github.com/DavidLapous/multipers and will be merged as a module of the `Gudhi` library. We also provide pseudo-code in Appendix H.

### 4.1    Convergence rates

In this section, we study the convergence rate of S-CDRs with respect to the number of sampled points, when computed from specific bifiltrations. Similar to the single parameter persistence setting [16], these rates are derived from Theorem 1. Indeed, since concentration inequalities for multiparameter

---

[2]Several different approaches can be used for computing decompositions [1, 7]. In our experiments, we used `MMA` [29] (with a family of 1000 diagonal lines) because of its simplicity and rapidity.

persistence modules have already been described in the literature, these concentration inequalities can transfer to our representations. Note that while Equations (7) and (8), which provide such rates, are stated for the S-CDR in (3), they also hold for the S-CDR in (2).

**Measure bifiltration.** Let $\mu$ be a compactly supported probability measure of $\mathbb{R}^D$, and let $\mu_n$ be the discrete measure associated to a sampling of $n$ points from $\mu$. The *measure bifiltration* associated to $\mu$ and $\mu_n$ is defined as $\mathcal{F}_{r,t}^\mu := \{x \in \mathbb{R}^D : \mu(B(x,r)) \leq t\}$, where $B(x,r)$ denotes the Euclidean ball centered on $x$ with radius $r$. Now, let $\mathbb{M}$ and $\mathbb{M}_n$ be the multiparameter persistence modules obtained from applying the homology functor on top of the measure bifiltrations $\mathcal{F}^\mu$ and $\mathcal{F}^{\mu_n}$. These modules are known to enjoy the following stability result [2, Theorem 3.1, Proposition 2.23 (i)]: $d_\mathrm{I}(\mathbb{M}, \mathbb{M}_n) \leq d_\mathrm{Pr}(\mu, \mu_n) \leq \min(d_W^p(\mu, \mu_n)^{\frac{1}{2}}, d_W^p(\mu, \mu_n)^{\frac{p}{p+1}})$, where $d_W^p$ and $d_\mathrm{Pr}$ stand for the $p$-Wasserstein and Prokhorov distances between probability measures. Combining these inequalities with Theorem 1, then taking expectations and applying the concentration inequalities of the Wasserstein distance (see [26, Theorem 3.1] and [21, Theorem 1]) lead to:

$$\delta\mathbb{E}\left[\|V_{\infty,\delta}(\mathbb{M}) - V_{\infty,\delta}(\mathbb{M}_n)\|_\infty\right] \leq \left(c_{p,q}\mathbb{E}\left(|X|^q\right)n^{-\left(\frac{1}{2p \vee d}\right) \wedge \frac{1}{p} - \frac{1}{q}}\log^{\alpha/q} n\right)^{\frac{p}{p+1}}, \qquad (7)$$

where $\vee$ stands for maximum, $\alpha = 2$ if $2p = q = d$, $\alpha = 1$ if $d \neq 2p$ and $q = dp/(d-p) \wedge 2p$ or $q > d = 2p$ and $\alpha = 0$ otherwise, $c_{p,q}$ is a constant that depends on $p$ and $q$, and $X$ is a random variable of law $\mu$.

**Čech complex and density.** A limitation of the measure bifiltration is that it can be difficult to compute. Hence, we now focus on another, easier to compute bifiltration. Let $X$ be a smooth compact $d$-submanifold of $\mathbb{R}^D$ ($d \leq D$), and $\mu$ be a measure on $X$ with density $f$ with respect to the uniform measure on $X$. We now define the bifiltration $\mathcal{F}^{C,f}$ with:

$$\mathcal{F}_{u,v}^{C,f} := \text{Čech}(u) \cap f^{-1}([v, \infty)) = \left\{x \in \mathbb{R}^D : d(x, X) \leq u, f(x) \geq v\right\}.$$

Moreover, given a set $X_n$ of $n$ points sampled from $\mu$, we also consider the approximate bifiltration $\mathcal{F}^{C,f_n}$, where $f_n\colon X \to \mathbb{R}$ is an estimation of $f$ (such as, e.g., a kernel density estimator). Let $\mathbb{M}$ and $\mathbb{M}_n$ be the multiparameter persistence modules associated to $\mathcal{F}^{C,f}$ and $\mathcal{F}^{C,f_n}$. Then, the stability of the interleaving distance [27, Theorem 5.3] ensures:

$$d_\mathrm{I}(\mathbb{M}, \mathbb{M}_n) \leq \|f - f_n\|_\infty \vee d_H(X, X_n),$$

where $d_H$ stands for the Hausdorff distance. Moreover, concentration inequalities for the Hausdorff distance and kernel density estimators are also available in the literature (see [16, Theorem 4] and [23, Corollary 15]). More precisely, when the density $f$ is $L$-Lipschitz and bounded from above and from below, i.e., when $0 < f_\mathrm{min} \leq f \leq f_\mathrm{max} < \infty$, and when $f_n$ is a kernel density estimator of $f$ with associated kernel $k$, one has:

$$\mathbb{E}(d_H(X, X_n)) \lesssim \left(\frac{\log n}{n}\right)^{\frac{1}{d}} \text{ and } \mathbb{E}(\|f - f_n\|_\infty) \lesssim Lh_n + \sqrt{\frac{\log(1/h_n)}{nh_n^d}},$$

where $h_n$ is the (adaptive) bandwidth of the kernel $k$. In particular, if $\mu$ is a measure comparable to the uniform measure of a $d = 2$-manifold, then for any stationary sequence $h_n := h > 0$, and considering a Gaussian kernel $k$, one has:

$$\delta\mathbb{E}\left[\|V_{\infty,\delta}(\mathbb{M}) - V_{\infty,\delta}(\mathbb{M}_n)\|_\infty\right] \lesssim \sqrt{\frac{\log n}{n}} + Lh. \qquad (8)$$

**Empirical convergence rates.** Now that we have established the theoretical convergence rates of S-CDRs, we estimate and validate them empirically on data sets. We will first study a synthetic data set and then a real data set of point clouds obtained with immunohistochemistry. We also illustrate how the stability of S-CDRs (stated in Theorem 1) is critical for obtaining such convergence in Appendix B, where we show that our main competitor, the multiparameter persistence image [11], is unstable and thus cannot achieve convergence, both theoretically and numerically.

*Annulus with non-uniform density.* In this synthetic example, we generate an annulus of 25,000 points in $\mathbb{R}^2$ with a non-uniform density, displayed in Figure 3a. Then, we compute the bifiltration $\mathcal{F}^{C,f_n}$ corresponding to the Alpha filtration and the sublevel set filtration of a kernel density estimator, with

bandwidth parameter $h = 0.1$, on the complete Alpha simplicial complex. Finally, we compute the candidate decompositions and associated S-CDRs of the associated multiparameter module (in homology dimension 1), and their normalized distances to the target representation, using either $\|\cdot\|_2^2$ or $\|\cdot\|_\infty$. The corresponding distances for various number of sample points are displayed in log-log plots in Figure 3b. One can see that the empirical rate is roughly consistent with the theoretical one ($-1/2$ for $\|\cdot\|_\infty$ and $-1$ for $\|\cdot\|_2$), even when $p \neq \infty$ (in which case our S-CDRs are stable for $d_\mathrm{B}$ but theoretically not for $d_\mathrm{I}$).

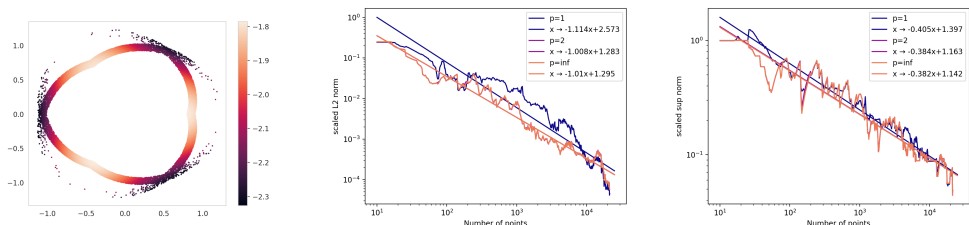

(a) Scatter plot of the synthetic data set colored by a kernel density estimator.

(b) **(left)** $\|\cdot\|_2^2$ and **(right)** $\|\cdot\|_\infty$ between the target representation and the empirical one w.r.t. $n$.

Figure 3: Convergence rate of synthetic data set.

*Immunohistochemistry data.* In our second experiment, we consider a point cloud representing cells, taken from [40], see Figure 4a. These cells are given with biological markers, which are typically used to assess, e.g., cell types and functions. In this experiment, we first triangulate the point cloud by placing a $100 \times 100$ grid on top of it. Then, we filter this grid using the sublevel set filtrations of kernel density estimators (with Gaussian kernel and bandwidth $h = 1$) associated to the CD8 and CD68 biological markers for immune cells. Finally, we compute the associated candidate decompositions of the multiparameter modules in homology dimensions 0 and 1, and we compute and concatenate their corresponding S-CDRs. Similar to the previous experiment, the theoretical convergence rate of our representations is upper bounded by the one for kernel density estimators with the $\infty$-norm. The convergence rates are displayed in Figure 4b. Again, one can see that the observed and theoretical convergence rates are consistent.

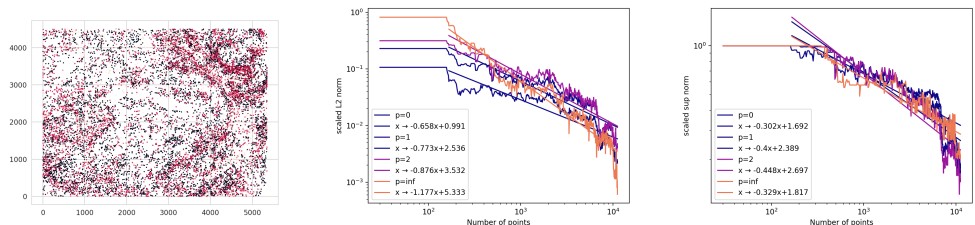

(a) Point cloud of cells colored by CD8 (red) and CD68 (black).

(b) **(left)** $\|\cdot\|_2^2$ and **(right)** $\|\cdot\|_\infty$ distances between the target representation and the empirical one w.r.t. $n$.

Figure 4: Convergence rate of immunohistochemistry data set.

## 4.2 Classification

In this section, we illustrate the efficiency of S-CDRs by using them for classification purposes. We show that they perform comparably or better than existing topological representations as well as standard baselines on several UCR benchmark data sets, graph data sets, and on the immunohisto-chemistry data set. Concerning UCR, we work with point clouds obtained from time delay embedding applied on the UCR time series, following the procedure of [10], and we produce S-CDRs with bifiltrations coming from combining either the Rips filtration with sublevel sets of a kernel density estimator (as in Section 4.1), or the Alpha filtration with the sublevel sets of the distance-to-measure

with parameter $m = 0.1$ (as in [10] and the baselines therein). Concerning graph datasets, we produce S-CDRs by filtering the graphs themselves directly using Heat Kernel Signature with parameter $t = 10$, Ricci curvature and node degree (similarly to what is used in the literature [11, 22, 45]).
In all tasks, every point cloud or graph has a label (corresponding to the type of its cells in the immunohistochemistry data set, and to pre-defined labels in the UCR and graph data sets), and our goal is to check whether we can predict these labels by training classifiers on the corresponding S-CDRs.

For point clouds (immunohistochemistry and UCR), we compare the performances of our S-CDRs (evaluated on a $50 \times 50$ grid) to the one of the multiparameter persistence landscape (MPL) [39], kernel (MPK) [17] and images (MPI) [10], as well as their single-parameter counterparts (P-L, P-I and PSS-K) [3]. We also compare to some non-topological baselines: we used the standard Ripley function evaluated on 100 evenly spaced samples in $[0, 1]$ for the immunohistochemistry data set, and k-NN classifiers with three difference distances for the UCR time series (denoted by B1, B2, B3), as suggested in [19]. For graps, we compare S-CDRs to the Euler characteristic based multiparameter persistence methods ECP, RT, and HTn, introduced in [22]. In order to also include non topological baselines, we also compare against the state-of-the-art graph classification methods RetGK [44], FGSD [37], and GIN [43].

All scores on the immunohistochemistry data set were computed after cross-validating a few classifiers (random forests, support vector machines and xgboost, with their default `Scikit-Learn` parameters) with 5 folds. For the time series data, our accuracy scores were obtained after also cross-validating the following S-CDR parameters; $p \in \{0, 1\}$, op $\in \{\text{sum}, \text{mean}\}$, $\delta \in \{0.01, 0.1, 0.5, 1\}$, $h \in \{0.1, 0.5, 1, 1.5\}$ with homology dimensions 0 and 1, and the following bandwidth parameters for kernel density esimation: $b \in \{0.1\%, 1\%, 10\%, 20\%, 30\%\}$, which are percentages of the diameters of the point clouds. with 5 folds. Parameters and results on graph data sets were cross-validated and averaged over 10 folds, following the pipelines of [22]. All results can be found in Table 1 (immunohistochemistry and UCR—UCR acronyms are provided in Appendix I) and Table 2. Bold indicates best accuracy and underline indicates best accuracy among topological methods. Note that there are no variances for UCR data sets since pre-defined train/test splits were provided. One can see that S-CDR almost always outperform topological baselines and are comparable to the standard baselines on the UCR benchmarks. Most notably, S-CDRs radically outperform the standard baseline and competing topological measures on the immunohistochemistry data set. For graph data sets, results are competitive with both topological and non-topological baselines; S-CDRs perform even slightly better on COX2.

| Dataset | B1 | B2 | B3 | PSS-K | P-I | P-L | MPK | MPL | MPI | S-CDR (Rips + KDE) | S-CDR (Alpha + DTM) |
|---|---|---|---|---|---|---|---|---|---|---|---|
| DPOAG | 62.6 | 62.6 | **77.0** | 76.9 | 69.8 | 70.5 | 67.6 | 70.5 | 71.9 | 71.9 | 71.9 |
| DPOC | 71.7 | 72.5 | 71.7 | 47.5 | 67.4 | 66.3 | **74.6** | 69.6 | 71.7 | 73.8 | **74.6** |
| PPOAG | 78.5 | 78.5 | 80.5 | 75.9 | 82.0 | 78.0 | 78.0 | 78.5 | 81.0 | 81.9 | **84.9** |
| PPOC | 80.8 | 79.0 | 78.4 | 78.4 | 72.2 | 72.5 | 78.7 | 78.7 | 81.8 | 79.4 | **83.2** |
| PPTW | 70.7 | 75.6 | 75.6 | 61.4 | 72.2 | 73.7 | 79.5 | 73.2 | 76.1 | 75.6 | 75.1 |
| IPD | **95.5** | **95.5** | 95.0 | - | 64.7 | 61.1 | 80.7 | 78.6 | 71.9 | 81.2 | 77.2 |
| GP | 91.3 | 91.3 | 90.7 | 90.6 | 84.7 | 80.0 | 88.7 | 94.0 | 90.7 | **96.3** | 92.7 |
| GPAS | 89.9 | **96.5** | 91.8 | - | 84.5 | 87.0 | 93.0 | 85.1 | 90.5 | 88.0 | 93.7 |
| GPMVF | 97.5 | 97.5 | **99.7** | - | 88.3 | 87.3 | 96.8 | 88.3 | 95.9 | 95.3 | 95.9 |
| PC | **93.3** | 92.2 | 87.8 | - | 83.4 | 76.7 | 85.6 | 84.4 | 86.7 | 93.1 | 90.0 |

| | Ripley | P | MPL | S-CDR |
|---|---|---|---|---|
| Immuno | 67.2(2.3) | 60.7(4.2) | 65.3(3.0) | **91.4(1.6)** |

Table 1: Accuracy scores for UCR and immunohistochemistry data sets.

---

[3]Note that the sizes of the point clouds in the immunohistochemistry data set were too large for MPK and MPI using the code provided in `https://github.com/MathieuCarriere/multipers`, and that all three single-parameter representations had roughly the same performance, so we only display one, denoted as P.

| Dataset | RetGK | FGSD | GIN | ECP | RT | HT nD | S-CDR |
|---|---|---|---|---|---|---|---|
| COX2 | 81.4(0.6) | - | - | 80.3(0.4) | 79.7(0.4) | 80.6(0.4) | **82.0(0.2)** |
| DHFR | 81.5(0.9) | - | - | 82.0(0.4) | 81.3(0.4) | **83.1(0.5)** | 81.6(0.2) |
| IMDB-B | 71.9(1.0) | 73.6 | **75.1(5.1)** | 73.3(0.4) | 74.0(0.5) | 74.7(0.5) | 73.5(0.2) |
| IMDB-M | 47.7(0.3) | **52.4** | 52.3(2.8) | 48.7(0.4) | 50.2(0.4) | 49.9(0.4) | 49.5(0.2) |
| MUTAG | 90.3(1.1) | **92.1** | 90(8.8) | 90.0(0.8) | 87.3(0.6) | 89.4(0.7) | 88.4(0.3) |
| PROTEINS | **78.0(0.3)** | 73.4 | 76.2(2.6) | 75.0(0.3) | 75.4(0.4) | 75.4(0.4) | 73.9(0.2) |

Table 2: Accuracy scores on graph datasets.

## 4.3 Running time comparisons

In this section, we provide running time comparisons between S-CDRs and the MPI and MPL representations, in which we measured the time needed to compute all the train and test S-CDRs and baselines of the previous data sets, averaged over the folds (again, note that since UCR data sets already provide the train/test splits, there is no variance in the corresponding results). All representations are evaluated on grids of sizes $50 \times 50$ and $100 \times 100$, and we provide the maximum running time over $p \in \{0, 1, \infty\}$. All computations were done using a Ryzen 4800 laptop CPU, with 16GB of RAM. We provide results in Table 3, where it can be seen that S-CDRs (computed on the pinched annulus and immunohistochemistry data sets) can be computed much faster than the other representations, by a factor of at least 25. As for UCR data sets, which contain only small time series and corresponding point clouds, it can still be observed that S-CDRs can be computed faster than the baselines. Interestingly, this sparse and fast implementation based on corners can also be used to improve on the running time of the multiparameter persistence landscapes (MPL), as one can see from Algorithm 4 in Appendix H (which retrieves the persistence barcode of a multiparameter persistence module along a given line; this is enough to compute the MPL) and from Table 3.

| | Annulus | Immuno | PPTW | GP |
|---|---|---|---|---|
| Ours (S-CDR) | $250ms(2ms)$ | $275ms(9.8ms)$ | $33.0ms(3.99ms)$ | $45.6ms(5.74ms)$ |
| Ours (MPL) | $36.9ms(0.8ms)$ | $65.9ms(0.9ms)$ | $22.4ms(2.15ms)$ | $31.8ms(2.95ms)$ |
| MPI (50) | $6.43s(25ms)$ | $5.67s(23.3ms)$ | $65.2ms(12.9ms)$ | $208ms(16.3ms)$ |
| MPL (50) | $17s(39ms)$ | $15.6s(14ms)$ | $154ms(27.9ms)$ | $630ms(30.0ms)$ |
| MPI (100) | $13.1s(125ms)$ | $11.65s(7.9ms)$ | $289ms(75.0ms)$ | $1.69s(77.7ms)$ |
| MPL (100) | $35s(193ms)$ | $31.3s(23.3ms)$ | $843ms(200ms)$ | $4.43s(186ms)$ |

Table 3: Running times for S-CDRs and competitors.

## 5 Conclusion

In this article, we study the general question of representing decompositions of multiparameter persistence modules in Topological Data Analysis. We first introduce T-CDR: a general template framework including specific representations (called S-CDR) that are provably stable. Our experiments show that S-CDR is superior to the state of the art.

**Limitations.** (1) Our current T-CDR parameter selection is currently done through cross-validation, which can be very time consuming and limits the number of parameters to choose from. (2) Our classification experiments were mostly illustrative. In particular, it would be useful to investigate more thoroughly on the influence of the T-CDR and S-CDR parameters, as well as the number of filtrations, on the classification scores. (3) In order to generate finite-dimensional vectors, we evaluated T-CDR and S-CDR on finite grids, which limited their discriminative powers when fine grids were too costly to compute.

**Future work.** (1) Since T-CDR is similar to the PersLay framework of single parameter persistence [11] and since, in this work, each of the framework parameter was optimized by a neural network, it is thus natural to investigate whether one can optimize T-CDR parameters in a data-driven way as well, so as to be able to avoid cross-validation. (2) In our numerical applications, we focused on representations computed off of MMA decompositions [29]. In the future, we plan to investigate whether working with other decomposition methods [1, 7] lead to better numerical performance when combined with our representation framework.

**Acknowledgments.** The authors would like to thank the area chair and anonymous reviewers for their insightful comments and constructive suggestions. The authors would also like to thank Hannah Schreiber for her great help with the implementation of our method. The authors are grateful to the OPAL infrastructure from Université Côte d'Azur for providing resources and support. DL was supported by ANR grant 3IA Côte d'Azur (ANR-19-P3IA-0002). MC was supported by ANR grant TopModel (ANR-23-CE23-0014). AJB was partially supported by ONR grant N00014-22-1-2679 and NSF grant DMS-2311338.

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

## A  Limitation of single-parameter persistent homology

A standard example of single-parameter filtration for point clouds, called the *Čech filtration*, is to consider $f : x \mapsto d_P(x) := \min_{p \in P} \|x - p\|$, where $P$ is a point cloud. The sublevel sets of this function are balls centered on the points in $P$ with growing radii, and the corresponding persistence barcode contains the topological features formed by $P$. See Figure 5a. When the radius of the balls is small, they form three connected components (**upper left**), identified as the three long red bars in the barcode (**lower**). When this radius is moderate, a cycle is formed (**upper middle**), identified as the long blue bar in the barcode (**lower**). Finally, when the radius is large, the balls cover the whole Euclidean plane, and no topological features are left, except for one connected component, that never dies (**upper right**).

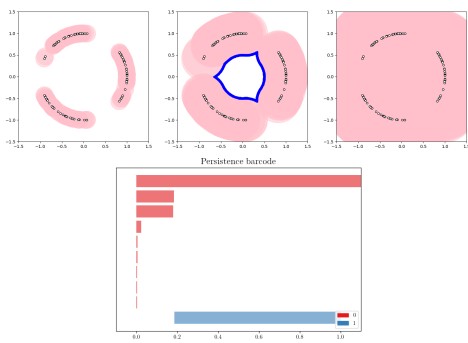 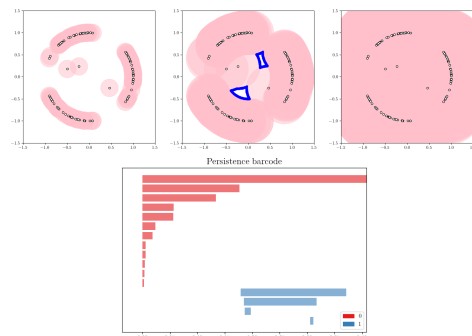

(a) Persistence barcode obtained from growing balls on a clean point cloud.

(b) Persistence barcode obtained from growing balls on a noisy point cloud.

Figure 5: Example of persistence barcode construction for point clouds using Čech filtration. In the middle sublevel sets, we highlight the topological cycles in blue.

The limitation of the Čech filtration alone for noisy point cloud is illustrated in Figure 5b, where only feature scale is used, and the presence of three outlier points messes up the persistence barcode entirely, since they induce the appearance of two cycles instead of one. In order to remedy this, one could remove points whose density is smaller than a given threshold, and then process the trimmed point cloud as before, but this requires using an arbitrary threshold choice to decide which points should be considered outliers or not.

## B  Unstability of MPI

In this section, we provide theoretical and experimental evidence that the multiparameter persistence image (MPI) [10], which is another decomposition-based representation from the literature, suffers from lack of stability as it does not enjoy guarantees such as the ones of S-CDRs (see Theorem 1). There are two main sources of instability in the MPI. The first one is due to the discretization induced by the lines: since it is obtained by placing Gaussian functions on top of slices of the intervals in the decompositions, if the Gaussian bandwidth $\sigma$ (see previous work, third item in Section 3.1) is too small w.r.t. the distance between consecutive lines in $L$, discretization effects can arise, as illustrated in Figure 6, in which the intervals do not appear as continuous shapes.

The second problem comes from the weight function: it is easy to build decompositions that are close in the interleaving distance, yet whose interval volumes are very different. See [5, Figure 15], and Figure 7, in which we show a family of modules $\mathbb{M}^\epsilon$ (**left**) made of a single interval built from connecting two squares with a bridge of diameter $\epsilon > 0$. We also show the distances between $\mathbb{M}^\epsilon$ and the limit module $\mathbb{M}$ made of two squares with no bridge, through their MPI and S-CDRs (**right**). Even though the interleaving distance between $\mathbb{M}^\epsilon$ and $\mathbb{M}$ goes to zero as $\epsilon$ goes to zero, the distances between MPI representations converge to a positive constant, whereas the distances between S-CDRs goes effectively to zero.

We also show that this lack of stability can even prevent convergence. In Figure 8, we repeated the setup of Section 4.1 on a synthetic data set sampled from a noisy annulus in the Euclidean plane. As

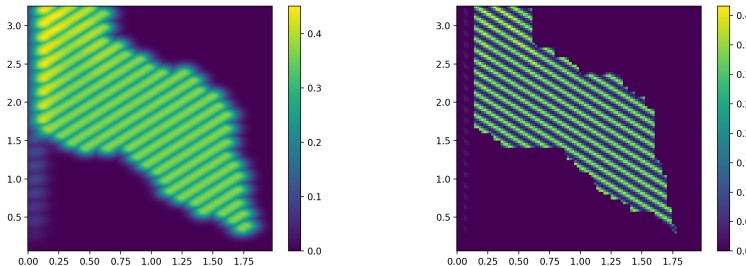

Figure 6: Examples of numerical artifacts occurring when the number of lines is too small w.r.t. the Gaussian bandwidth, for two different families of lines.

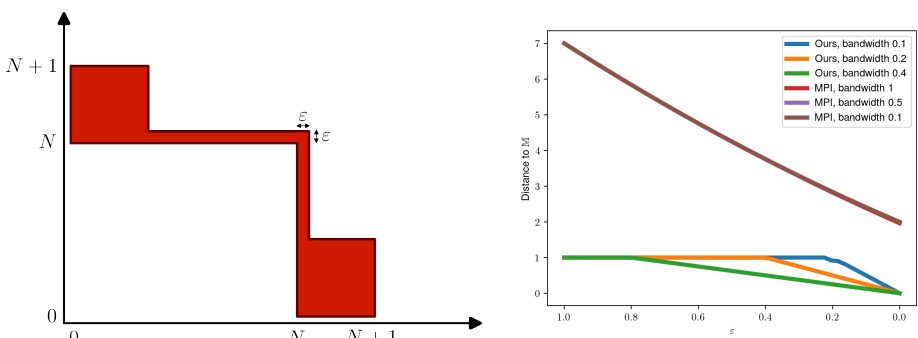

Figure 7: Example of unstability of MPI.

one can clearly see, convergence does not happen for MPI as the Euclidean distance to the limit MPI representation associated to the maximal number of subsample points suddenly drops to zero after an erratic behavior, while S-CDRs exhibit a gradual decrease in agreement with Theorem 1.

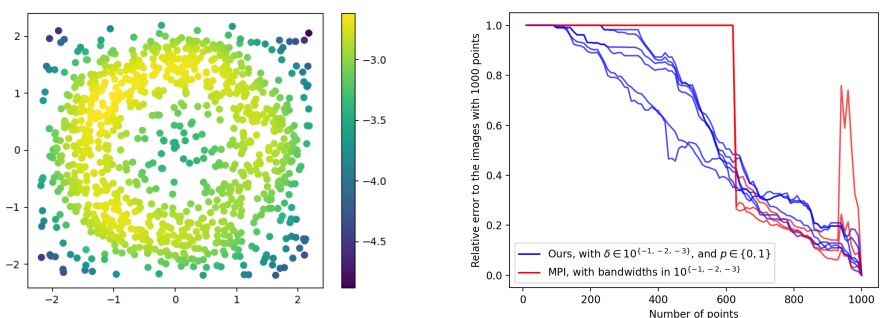

Figure 8: Example of lack of convergence for MPI.

## C    Simplicial homology

In this section, we recall the basics of simplicial homology with coefficients in $\mathbb{Z}/2\mathbb{Z}$, which we use for practical computations. A more detailed presentation can be found in [30, Chapter 1]. The basic bricks of simplicial (persistent) homology are *simplicial complexes*, which are combinatorial models of topological spaces that can be stored and processed numerically.

**Definition 3.** Given a set of points $X_n := \{x_1, \ldots, x_n\}$ sampled in a topological space $X$, an *abstract simplicial complex* built from $X_n$ is a family $S(X_n)$ of subsets of $X_n$ such that:

- if $\tau \in S(X_n)$ and $\sigma \subseteq \tau$, then $\sigma \in S(X_n)$, and

- if $\sigma, \tau \in S(X_n)$, then either $\sigma \cap \tau \in S(X_n)$ or $\sigma \cap \tau = \varnothing$.

Each element $\sigma \in S(X_n)$ is called a *simplex* of $S(X_n)$, and the *dimension* of a simplex is defined as $\dim(\sigma) := \mathrm{card}(\sigma) - 1$. Simplices of dimension 0 are called *vertices*.

An important linear operator on simplices is the so-called *boundary operator*. Roughly speaking, it turns a simplex into the *chain* of its faces, where a chain is a formal sum of simplices. The set of chains has a group structure, denoted by $Z_*(S(X_n))$.

**Definition 4.** Given a simplex $\sigma := [x_{i_1}, \ldots, x_{i_p}]$, the boundary operator $\partial$ is defined as

$$\partial(\sigma) := \sum_{j=1}^{p} [x_{i_1}, \ldots, x_{i_{j-1}}, x_{i_{j+1}}, \ldots, x_{i_p}].$$

In other words, it is the chain constructed from $\sigma$ by removing one vertex at a time. This operator $\partial$ can then be extended straightforwardly to chains by linearity.

Given a simplicial complex, a topological feature is defined as a *cycle*, i.e., a chain such that each simplex in its boundary appears an even number of times. In order to formalize this property, we remove a simplex in a chain every time it appears twice, and we let a cycle be a chain $c$ s.t. $\partial(c) = 0$.

Now, one can easily check that $\partial \circ \partial(c) = 0$ for any chain $c$, i.e., the boundary of a cycle is always a cycle. Hence, one wants to exclude cycles that are boundaries, since they correspond somehow to trivial cycles. Again, such boundaries form a group that is denoted by $B_*(S(X_n))$.

**Definition 5.** The homology group in dimension $k$ is the quotient group

$$H_k(S(X_n)) := \frac{Z_k(S(X_n))}{B_k(S(X_n))}.$$

In other words, it is the group (one can actually show it is a vector space) of cycles made of simplices of dimension $k$ that are not boundaries.

See Figure 9 for an illustration of these definitions. Finally, given a filtered simplicial complex (with a filtration defined as in Section 2), computing its associated persistence barcode using the simplicial homology functor can be done with several softwares, such as, e.g., Gudhi [36].

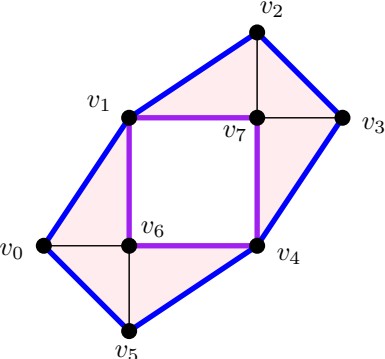

Figure 9: Example of simplicial complex $S$ built from eight points, and made of eight vertices (simplices of dimension 0), fourteen edges (simplices of dimension 1) and six triangles (simplices of dimension 2). The purple path is a cycle; indeed $\partial([v_1, v_7] + [v_7, v_4] + [v_4, v_6] + [v_6, v_1]) = [v_1] + [v_7] + [v_7] + [v_4] + [v_4] + [v_6] + [v_6] + [v_1] = 0$ since every vertex appears twice. Similarly, the blue path is a cycle as well. However, both paths represent the same topological feature, in the sense that they belong to the same equivalence class of $H_1(S)$, since their sum is exactly the boundary of the 2-chain comprised of the six triangles of the complex, i.e., they differ only by a trivial cycle. Hence, the dimension of $H_1(S)$ is 1.

# D  Modules, interleaving and bottleneck distances

In this section, we provide a more formalized version of multiparameter persistence modules and their associated distances. Strictly speaking, multiparameter persistence modules are nothing but a parametrized family of vector spaces obtained from applying the homology functor to a multifiltration, as explained in Section 2.

**Definition 6.** A *multiparameter persistence module* $\mathbb{M}$ is a family of vector spaces $\{M(\alpha) : \alpha \in \mathbb{R}^n\}$, together with linear transformations, also called *transition maps*, $\varphi_\alpha^\beta : \mathbb{M}(\alpha) \to \mathbb{M}(\beta)$ for any $\alpha \leq \beta$ (where $\leq$ denotes the partial order of $\mathbb{R}^n$), that satisfy $\varphi_\alpha^\gamma = \varphi_\beta^\gamma \circ \varphi_\alpha^\beta$ for any $\alpha \leq \beta \leq \gamma$.

Of particular interest are *interval modules*, since they are easier to work with.

**Definition 7.** An *interval module* $\mathbb{M}$ is a multiparameter persistence module such that:

- its dimension is at most 1: $\dim(\mathbb{M}(\alpha)) \leq 1$ for any $\alpha \in \mathbb{R}^n$, and

- its support $\mathrm{supp}(\mathbb{M}) := \{\alpha \in \mathbb{R}^n : \dim(\mathbb{M}(\alpha)) = 1\}$ is an interval of $\mathbb{R}^n$,

where an interval of $\mathbb{R}^n$ is a subset of $I \subseteq \mathbb{R}^n$ that satisfy:

- *(convexity)* if $p, q \in I$ and $p \leq r \leq q$ then $r \in I$, and

- *(connectivity)* if $p, q \in I$, then there exists a finite sequence $r_1, r_2, \ldots, r_m \in I$, for some $m \in \mathbb{N}$, such that $p \sim r_1 \sim r_2 \sim \cdots \sim r_m \sim q$, where $\sim$ can be either $\leq$ or $\geq$.

In the main body of this article, we study representations for *candidate decompositions* of modules, i.e., direct sums[4] of interval modules that approximate the original modules.

Multiparameter persistence modules can be compared with the *interleaving distance* [27].

**Definition 8** (Interleaving distance). Given $\varepsilon > 0$, two multiparameter persistence modules $\mathbb{M}$ and $\mathbb{M}'$ are $\varepsilon$-*interleaved* if there exist two morphisms $f \colon \mathbb{M} \to \mathbb{M}'_\varepsilon$ and $g \colon \mathbb{M}' \to \mathbb{M}_\varepsilon$ such that $g_{\cdot+\varepsilon} \circ f_\cdot = \varphi_\cdot^{+2\varepsilon}$ and $f_{\cdot+\varepsilon} \circ g_\cdot = \psi_\cdot^{+2\varepsilon}$, where $\mathbb{M}_\varepsilon$ is the *shifted module* $\{\mathbb{M}(x + \varepsilon)\}_{x \in \mathbb{R}^n}$, $\varepsilon = (\varepsilon, \ldots, \varepsilon) \in \mathbb{R}^n$, and $\varphi$ and $\psi$ are the transition maps of $\mathbb{M}$ and $\mathbb{M}'$ respectively. The *interleaving distance* between two multiparameter persistence modules $\mathbb{M}$ and $\mathbb{M}'$ is then defined as $d_\mathrm{I}(\mathbb{M}, \mathbb{M}') := \inf \{\varepsilon \geq 0 : \mathbb{M}$ and $\mathbb{M}'$ are $\varepsilon$-interleaved$\}$.

The main property of this distance is that it is *stable* for multi-filtrations that are obtained from the sublevel sets of functions. More precisely, given two continuous functions $f, g : S \to \mathbb{R}^n$ defined on a simplicial complex $S$, let $\mathbb{M}(f), \mathbb{M}(g)$ denote the multiparameter persistence modules obtained from the corresponding multifiltrations $\{S_x^f := \{\sigma \in S : f(\sigma) \leq x\}\}_{x \in \mathbb{R}^n}$ and $\{S_x^g := \{\sigma \in S : g(\sigma) \leq x\}\}_{x \in \mathbb{R}^n}$. Then, one has [27, Theorem 5.3]:

$$d_\mathrm{I}(\mathbb{M}(f), \mathbb{M}(g)) \leq \|f - g\|_\infty. \tag{9}$$

Another usual distance is the *bottleneck distance* [5, Section 2.3]. Intuitively, it relies on decompositions of the modules into direct sums of indecomposable summands[5] (which are not necessarily intervals), and is defined as the largest interleaving distance between summands that are matched under some matching.

**Definition 9** (Bottleneck distance). Given two multisets $A$ and $B$, $\mu \colon A \nrightarrow B$ is called a *matching* if there exist $A' \subseteq A$ and $B' \subseteq B$ such that $\mu \colon A' \to B'$ is a bijection. The subset $A' := \mathrm{coim}(\mu)$ (resp. $B' := \mathrm{im}(\mu)$) is called the *coimage* (resp. *image*) of $\mu$.

Let $\mathbb{M} \cong \bigoplus_{i \in \mathcal{I}} M_i$ and $\mathbb{M}' \cong \bigoplus_{j \in \mathcal{J}} M_j'$ be two multiparameter persistence modules. Given $\varepsilon \geq 0$, the modules $\mathbb{M}$ and $\mathbb{M}'$ are $\varepsilon$-*matched* if there exists a matching $\mu \colon \mathcal{I} \nrightarrow \mathcal{J}$ such that $M_i$ and $M'_{\mu(i)}$ are $\varepsilon$-interleaved for all $i \in \mathrm{coim}(\mu)$, and $M_i$ (resp. $M_j'$) is $\varepsilon$-interleaved with the null module $\mathbf{0}$ for all $i \in \mathcal{I} \backslash \mathrm{coim}(\mu)$ (resp. $j \in \mathcal{J} \backslash \mathrm{im}(\mu)$).

The *bottleneck distance* between two multiparameter persistence modules $\mathbb{M}$ and $\mathbb{M}'$ is then defined as $d_\mathrm{B}(\mathbb{M}, \mathbb{M}') := \inf \{\varepsilon \geq 0 : \mathbb{M}$ and $\mathbb{M}'$ are $\varepsilon$-matched$\}$.

---

[4]Since multiparameter persistence modules are essentially families of vector spaces connected by transition maps, they admit direct sum decompositions, pretty much like usual vector spaces do.

[5]Recall that a module $\mathbb{M}$ is *indecomposable* if $\mathbb{M} \cong A \oplus B \Rightarrow \mathbb{M} \simeq A$ or $\mathbb{M} \simeq B$.

Since a matching between the decompositions of two multiparameter persistence modules induces an interleaving between the modules themselves, it follows that $d_\mathrm{I} \leq d_\mathrm{B}$. Note also that $d_\mathrm{B}$ can actually be arbitrarily larger than $d_\mathrm{I}$, as showcased in [5, Section 9].

# E   The fibered barcode and its properties

**Definition 10.** Let $n \in \mathbb{N}^*$ and $F = \{F^{(1)}, \ldots, F^{(n)}\}$ be a multifiltration on a topological space $X$. Let $\mathbf{e}, b \in \mathbb{R}^n$, and $\ell_{\mathbf{e},b} : \mathbb{R} \to \mathbb{R}^n$ be the line in $\mathbb{R}^n$ defined with $\ell_{\mathbf{e},b}(t) = t \cdot \mathbf{e} + b$, that is, $\ell_{\mathbf{e},b}$ is the line of direction $\mathbf{e}$ passing through $b$. Let $F_{\mathbf{e},b} : \mathbb{R} \to \mathcal{P}(X)$ defined with $F_{\mathbf{e},b}(t) = \bigcap_{i=1}^n F^{(i)}([\ell_{\mathbf{e},b}(t)]_i)$, where $[\cdot]_i$ denotes the $i$-th coordinate. Then, each $F_{\mathbf{e},b}$ is a single-parameter filtration and has a corresponding persistence barcode $B_{\mathbf{e},b}$. The set $\mathcal{B}(F) = \{B_{\mathbf{e},b} : \mathbf{e}, b \in \mathbb{R}^n\}$ is called the *fibered barcode* of $F$.

The two following lemmas from [25] describe two useful properties of the fibered barcode.

**Lemma 1** (Lemma 1 in [25])**.** *Let $\mathbf{e}, b \in \mathbb{R}^n$ and $\ell_{\mathbf{e},b}$ be the corresponding line. Let $\hat{e} = \min_i[\mathbf{e}]_i$. Let $F, F'$ be two multi-filtrations, $\mathbb{M}, \mathbb{M}'$ be the corresponding persistence modules and $B_{\mathbf{e},b} \in \mathcal{B}(F)$ and $B'_{\mathbf{e},b} \in \mathcal{B}(F')$ be the corresponding barcodes in the fibered barcodes of $F$ and $F'$. Then, the following stability property holds:*

$$d_\mathrm{B}(B_{\mathbf{e},b}, B'_{\mathbf{e},b}) \leq \frac{d_\mathrm{I}(\mathbb{M}, \mathbb{M}')}{\hat{e}}. \tag{10}$$

**Lemma 2** (Lemma 2 in [25])**.** *Let $\mathbf{e}, \mathbf{e}', b, b' \in \mathbb{R}^n$ and $\ell_{\mathbf{e},b}, \ell_{\mathbf{e}',b'}$ be the corresponding lines. Let $\hat{e} = \min_i[\mathbf{e}]_i$ and $\hat{e}' = \min_i[\mathbf{e}']_i$. Let $F$ be a multi-filtration, $\mathbb{M}$ be the corresponding persistence module and $B_{\mathbf{e},b}, B_{\mathbf{e}',b'} \in \mathcal{B}(F)$ be the corresponding barcodes in the fibered barcode of $F$. Assume $\mathbb{M}$ is decomposable $\mathbb{M} = \oplus_{i=1}^m M_i$, and let $K > 0$ such that $M_i \subseteq B_\infty(0, K) := \{x \in \mathbb{R}^n : \|x\|_\infty \leq K\}$ for all $i \in [\![1, m]\!]$. Then, the following stability property holds:*

$$d_\mathrm{B}(B_{\mathbf{e},b}, B_{\mathbf{e}',b'}) \leq \frac{(K + \max\{\|b\|_\infty, \|b'\|_\infty\}) \cdot \|\mathbf{e} - \mathbf{e}'\|_\infty + \|b - b'\|_\infty}{\hat{e} \cdot \hat{e}'}. \tag{11}$$

# F   Proof of Theorem 1

Our proof is based on several lemmas. In the first one, we focus on the S-CDR weight function $w$ as defined in Definition 2.

**Lemma 3.** *Let $M$ and $M'$ be two interval modules with compact support. Then, one has*

$$d_\mathrm{I}(M, 0) = \frac{1}{2} \sup_{b,d \in \mathrm{supp}(M)} \min_j (d_j - b_j)_+ = w(M). \tag{12}$$

*Furthermore, one has the equality*

$$|w(M) - w(M')| \leq d_\mathrm{I}(M, M'). \tag{13}$$

*Proof.* We first show Equation (12) with two inequalities.

**First inequality:** $\leq$ Let $M$ be an interval module. If $d_\mathrm{I}(M, 0) = 0$, then the inequality is trivial, so we now assume that $d_\mathrm{I}(M, 0) > 0$. Let $\delta > 0$ such that $\delta < d_\mathrm{I}(M, 0)$. By definition of $d_\mathrm{I}$, the identity morphism $M \to M_{2\delta}$ cannot be factorized by 0. This implies the existence of some $b \in \mathbb{R}^n$ such that $\mathrm{rank}(M(b) \to M(b + 2\boldsymbol{\delta})) > 0$; in particular, $b, b + 2\boldsymbol{\delta} \in \mathrm{supp}(M)$. Making $\delta$ converge to $d_\mathrm{I}(M, 0)$ yields the desired inequality.

**Second inequality:** $\geq$ Let $(K_n)_{n \in \mathbb{N}}$ be a compact interval exhaustion of $\mathrm{supp}(M)$, and $b_n, d_n \in K_n$ be two points that achieve the maximum in

$$\frac{1}{2} \sup_{b,d \in K_n} \min_j (d_j - b_j)_+.$$

Now, by functoriality of persistence modules, we can assume without loss of generality that $b_n$ and $d_n$ are on the same diagonal line (indeed, if they are not, it is possible to transform $d_n$ into

$\tilde{d}_n$ such that $b_n$ and $\tilde{d}_n$ are on the same diagonal line and also achieve the supremum). Thus, $\operatorname{rank}(M(b_n) \to M(d_n)) > 0$, and $d_{\mathrm{I}}(M, 0) \geq \frac{1}{2}\|d_n - b_n\|_\infty$. Taking the limit over $n \in \mathbb{N}$ leads to the desired inequality.

Inequality (13) follows directly from the triangle inequality applied on $d_{\mathrm{I}}$.

$\square$

In the following lemma, we rewrite volumes of interval module supports using interleaving distances.

**Lemma 4.** *Let $M$ be an interval module, and $R \subseteq \mathbb{R}^n$ be a compact rectangle, with $n \geq 2$. Then, one has:*
$$\operatorname{vol}\left(\operatorname{supp}\left(M\right) \cap R\right) = 2 \int_{\{y \in \mathbb{R}^n : y_n = 0\}} d_{\mathrm{I}}\left(M\big|_{l_y \cap R}, 0\right) \, \mathrm{d}\lambda^{n-1}(y)$$
*where $l_y$ is the diagonal line crossing $y$, and $\lambda^{n-1}$ denotes the Lebesgue measure in $\mathbb{R}^{n-1}$.*

*Proof.* Using the change of variables $y_i = x_i - x_n$ and $t = x_n$ (which has a trivial Jacobian) yields the following inequalities:
$$\operatorname{vol}(\operatorname{supp}\left(M\right) \cap R) = \int_{\operatorname{supp}(M) \cap R} \mathrm{d}\lambda^n(x)$$
$$= \int_{\{y \in \mathbb{R}^n : y_n = 0\}} \int_{t \in \mathbb{R}} \mathbb{1}_{\operatorname{supp}(M) \cap R}(y + t) \, \mathrm{d}t \, \mathrm{d}\lambda^{n-1}(y)$$
$$= \int_{\{y \in \mathbb{R}^n : y_n = 0\}} \operatorname{diam}_{\|\cdot\|_\infty}\left(\operatorname{supp}\left(M\right) \cap l_y \cap R\right) \mathrm{d}\lambda^{n-1}(y)$$
where $l_y$ is the diagonal line passing through $y$. Now, since $M$ is an interval module, one has $\operatorname{diam}_{\|\cdot\|_\infty}\left(\operatorname{supp}\left(M\right) \cap l_y \cap R\right) = 2d_{\mathrm{I}}(M\big|_{l_y \cap R}, 0)$, which concludes the proof.

$\square$

In the following proposition, we provide stability bounds for single interval modules.

**Proposition 1.** *If $M$ and $M'$ are two interval modules, then for any $\delta > 0$ and S-CDR parameter $\phi_\delta$ in Definition 2, one has:*

1. *$0 \leq \phi_\delta(M)(x) \leq \frac{w(M)}{\delta} \wedge 1$, for any $x \in \mathbb{R}^n$,*

2. *$\|\phi_\delta(M) - \phi_\delta(M')\|_\infty \leq 2(d_{\mathrm{I}}(M, M') \wedge \delta)/\delta$.*

*Proof.* Claim 1. is a simple consequence of Equation (12).

Claim 2. for S-CDR parameter (a) is a simple consequence of the triangle inequality.

Let us prove Claim 2. for (b). Let $x \in \mathbb{R}^n$ and $\delta > 0$. One has:
$$|\phi_\delta(M)(x) - \phi_\delta(M')(x)| \leq \frac{2}{(2\delta)^n} \int_{\{y : y_n = 0\}} |d_{\mathrm{I}}\left(M\big|_{l_y \cap R_{x,\delta}}, 0\right) - d_{\mathrm{I}}\left(M'\big|_{l_y \cap R_{x,\delta}}, 0\right)| \, \mathrm{d}\lambda^{n-1}(y)$$
$$\leq \frac{2}{(2\delta)^n} \int_{\{y : y_n = 0\}} d_{\mathrm{I}}\left(M\big|_{l_y \cap R_{x,\delta}}, M'\big|_{l_y \cap R_{x,\delta}}\right) \mathrm{d}\lambda^{n-1}(y)$$
$$\leq 2(d_{\mathrm{I}}(M, M') \wedge \delta)/\delta,$$
where the first inequality comes from Lemma 4, the second inequality is an application of the triangle inequality, and the third inequality comes from Lemma 1.

Finally, let us prove Claim 2. for (c). Let $x \in \mathbb{R}^n$ and $\delta > 0$. Let $b \leq d \in \operatorname{supp}\left(M\right) \cap R_{x,\delta}$. Let also $\gamma > 0$. Then, using Lemma 4, one has:
$$\frac{1}{(2\delta)^n} \operatorname{vol}(\operatorname{supp}\left(M\right) \cap R_{b,d}) = \frac{2}{(2\delta)^n} \int_{\{y \in \mathbb{R}^n : y_n = 0\}} d_{\mathrm{I}}(M\big|_{R_{b,d} \cap l_y}, 0) \, \mathrm{d}\lambda^{n-1}(y)$$
$$\leq \frac{2}{\delta}\gamma + \frac{2}{(2\delta)^n} \int_{\{y \in \mathbb{R}^n : y_n = 0\}} d_{\mathrm{I}}(M\big|_{R_{b+\gamma,d-\gamma} \cap l_y}, 0) \, \mathrm{d}\lambda^{n-1}(y),$$

using the convention $R_{a,b} = \varnothing$ if $a \not\leq b$. Now, set $\gamma := d_{\mathrm{I}}(M\big|_{R_{x,\delta}}, M'\big|_{R_{x,\delta}})$. If $b + \boldsymbol{\gamma}$ or $d - \boldsymbol{\gamma} \notin \mathrm{supp}\,(M')$ then $d_{\mathrm{I}}(M\big|_{R_{x,\delta}}, M'\big|_{R_{x,\delta}}) = \gamma > d_{\mathrm{I}}(M, M')$ which is impossible. Thus,

$$
\frac{1}{(2\delta)^n}\mathrm{vol}(R_{b,d}) \leq 2d_{\mathrm{I}}(M\big|_{R_{x,\delta}}, M'\big|_{R_{x,\delta}})/\delta
$$
$$
+ \sup_{a,c \in R_{x,\delta} \cap \mathrm{supp}(M')} \frac{2}{(2\delta)^n} \int_{\{y \in \mathbb{R}^n : y_n = 0\}} d_{\mathrm{I}}(M\big|_{R_{a,c} \cap l_y}, 0)\,\mathrm{d}\lambda^{n-1}(y)
$$
$$
= 2d_{\mathrm{I}}(M\big|_{R_{x,\delta}}, M'\big|_{R_{x,\delta}})/\delta + \phi_\delta(M')(x)
$$

Finally, taking the supremum on $b \leq d \in \mathrm{supp}\,(M) \cap R_{x,\delta}$ yields

$$
\phi_\delta(M)(x) - \phi_\delta(M')(x) \leq 2d_{\mathrm{I}}(M\big|_{R_{x,\delta}}, M'\big|_{R_{x,\delta}})/\delta \leq 2\left(d_{\mathrm{I}}(M, M') \wedge \delta\right)/\delta.
$$

The desired inequality follows by symmetry on $M$ and $M'$.

$\square$

Equipped with these results, we can finally prove Theorem 1.

*Proof. Theorem 1.*

Let $\mathbb{M} = \oplus_{i=1}^m M_i$ and $\mathbb{M}' = \oplus_{j=1}^{m'} M_j'$ be two modules that are decomposable into interval modules and $x \in \mathbb{R}^n$.

**Inequality 5.** To simplify notations, we define the following: $w_i := w(M_i)$, $\phi_{i,x} := \phi_\delta(M_i)(x)$ and $w_j' := w(M_j')$, $\phi_{j,x}' := \phi_\delta(x, M_j')$. Let us also assume without loss of generality that the indices are consistent with a matching achieving the bottleneck distance. In other words, the bottleneck distance is achieved for a matching that matches $M_i$ with $M_i'$ for every $i$ (up to adding 0 modules in the decompositions of $\mathbb{M}$ and $\mathbb{M}'$ so that $m = m'$). Finally, assume without loss of generality that $\sum_i w_i' \geq \sum_i w_i$. Then, one has:

$$
|V_{1,\delta}(\mathbb{M})(x) - V_{1,\delta}(\mathbb{M}')(x)| = \left| \frac{1}{\sum_i w_i} \sum_i w_i \phi_{i,x} - \frac{1}{\sum_i w_i'} \sum_i w_i' \phi_{i,x}' \right|
$$
$$
\leq \frac{1}{\sum_i w_i'} \left| \sum_i w_i \phi_{i,x} - \sum_i w_i' \phi_{i,x}' \right| + \left| \frac{1}{\sum_i w_i} - \frac{1}{\sum_i w_i'} \right| \left| \sum_i w_i \phi_{i,x} \right|.
$$

Now, for any index $i$, since $d_{\mathrm{I}}(M_i, M_i') \leq d_{\mathrm{B}}(\mathbb{M}, \mathbb{M}')$ and $|w_i - w_i'| \leq d_{\mathrm{I}}(M_i, M_i') \leq d_{\mathrm{B}}(\mathbb{M}, \mathbb{M}')$ by Lemma 3, Proposition 1 ensures that:

$$
|w_i \phi_{i,x} - w_i' \phi_{i,x}'| \leq |w_i - w_i'|\phi_{i,x} + w_i'|\phi_{i,x} - \phi_{i,x}'| \leq 2(w_i + w_i')(d_{\mathrm{B}}(\mathbb{M}, \mathbb{M}') \wedge \delta)/\delta
$$

and

$$
\left| \frac{1}{\sum_i w_i} - \frac{1}{\sum_i w_i'} \right| \leq \frac{1}{\sum_i w_i'} \left| \frac{\sum_i w_i' - w_i}{\sum_i w_i} \right| \leq \frac{m d_{\mathrm{B}}(\mathbb{M}, \mathbb{M}')}{(\sum_i w_i')(\sum_i w_i)}.
$$

Finally,

$$
|V_{1,\delta}(\mathbb{M})(x) - V_{1,\delta}(\mathbb{M}')(x)| \leq \left[ \frac{\sum_i w_i + w_i'}{\sum_i w_i'} + \frac{\sum_i w_i}{\frac{1}{m}(\sum_i w_i)(\sum_i w_i')} \right] 2(d_{\mathrm{B}}(\mathbb{M}, \mathbb{M}') \wedge \delta)/\delta
$$
$$
\leq \left[ 4 + \frac{2}{C} \right] (d_{\mathrm{B}}(\mathbb{M}, \mathbb{M}') \wedge \delta)/\delta.
$$

**Inequality 4** can be proved using the proof of Inequality 5 by replacing every $w_i$ by 1.

**Inequality 6.** Let us prove the inequality for (b). Let $R := R_{x-\delta,x+\delta}$. One has:

$$V_{\infty,\delta}(\mathbb{M})(x) - V_{\infty,\delta}(\mathbb{M}')(x) =$$

$$\sup_i \frac{2}{(2\delta)^n} \int_{\{y\in\mathbb{R}^n:y_n=0\}} d_{\mathrm{I}}(M_i|_{l_y\cap R},0)\,\mathrm{d}\lambda^{n-1}(y)$$

$$-\sup_j \frac{2}{(2\delta)^n} \int_{\{y\in\mathbb{R}^n:y_n=0\}} d_I(M'_j|_{l_y\cap R},0)\,\mathrm{d}\lambda^{n-1}(y)$$

$$\textit{(for any index j)} \leq \sup_i \frac{2}{(2\delta)^n} \int_{\{y\in\mathbb{R}^n:y_n=0\}} d_{\mathrm{I}}(M_i|_{l_y\cap R},0) - d_{\mathrm{I}}(M'_j|_{l_y\cap R},0)\,\mathrm{d}\lambda^{n-1}(y)$$

$$\leq \frac{2}{(2\delta)^n} \int_{\{y\in\mathbb{R}^n:y_n=0\}} \sup_i\inf_j d_{\mathrm{I}}(M_i|_{l_y\cap R},M'_j|_{l_y\cap R})\,\mathrm{d}\lambda^{n-1}(y)$$

Now, as the interleaving distance is equal to the bottleneck distance for single parameter persistence [15, Theorem 5.14], one has:

$$\sup_i\inf_j d_{\mathrm{I}}(M_i|_{l_y\cap R},M'_j|_{l_y\cap R}) \leq d_{\mathrm{B}}(\mathbb{M}|_{l_y\cap R},\mathbb{M}'|_{l_y\cap R}) = d_{\mathrm{I}}(\mathbb{M}|_{l_y\cap R},\mathbb{M}'|_{l_y\cap R}) \leq d_{\mathrm{I}}(\mathbb{M},\mathbb{M}')\wedge\delta$$

which leads to the desired inequality. The proofs for (a) and (c) follow the same lines (upper bound the suprema in the right hand term with either infima or appropriate choices in order to reduce to the single parameter case). $\qquad\square$

## G    An additional stability theorem

In this section, we define a new S-CDR, with a slightly different type of upper bound. It relies on the fibered barcode introduced in Appendix E. We also slightly abuse notations and use $M_i$ to denote both an interval module and its support.

**Proposition 2.** *Let $\mathbb{M} \simeq \oplus_{i=1}^m M_i$ be a multiparameter persistence module that can be decomposed into interval modules. Let $\sigma > 0$, and let $0 \leq \delta \leq \delta(\mathbb{M})$, where*

$$\delta(\mathbb{M}) := \inf\{\delta \geq 0 : \Gamma_{\mathbb{M}} \text{ achieves } d_{\mathrm{B}}(B_{\mathbf{e}_\Delta,x},B_{\mathbf{e}_\Delta,x+\delta\mathbf{u}}) \text{ for all } x,\mathbf{u} \text{ s.t. } \|\mathbf{u}\|_\infty = 1, \langle\mathbf{e}_\Delta,\mathbf{u}\rangle = 0\},$$

*where $\Gamma_{\mathbb{M}}$ is the partial matching induced by the decomposition of $\mathbb{M}$. Let $\mathcal{N}(x,\sigma)$ denote the function*

$$\mathcal{N}(x,\sigma): \begin{cases} \mathbb{R}^n & \to \mathbb{R} \\ p & \mapsto \exp\left(-\frac{\|p-x\|^2}{2\sigma^2}\right) \end{cases} \quad \textit{and let}$$

$$V_{\delta,\sigma}(\mathbb{M}): \begin{cases} \mathbb{R}^n & \to \mathbb{R} \\ x & \mapsto \max_{1\leq i\leq m}\ \max_{f\in\mathcal{C}(x,\delta,M_i)}\ \|\mathcal{N}(x,\sigma)\cdot f\|_1 \end{cases} \tag{14}$$

*where $\mathcal{C}(x,\delta,M_i)$ stands for the set of interval functions from $\mathbb{R}^n$ to $\{0,1\}$ whose support is $T_\delta(\ell) \cap M_i$, where $\ell$ is a connecting component of $\mathrm{im}(\ell_{\mathbf{e}_\Delta,x}) \cap M_i$ and $\mathbf{e}_\Delta = [1,\dots,1] \in \mathbb{R}^n$, and where $T_\delta(\ell)$ is the $\delta$-thickening of the line $L(\ell)$ induced by $\ell$: $T_\delta(\ell) = \{x \in \mathbb{R}^k : \|x,L(\ell)\|_\infty \leq \delta\}$.*

*Then, $V_{\delta,\sigma}$ satisfies the following stability property:*

$$\|V_{\delta,\sigma}(\mathbb{M}) - V_{\delta,\sigma}(\mathbb{M}')\|_\infty \leq (\sqrt{\pi}\sigma)^n \cdot \sqrt{2^{n+1}\delta^{n-1}d_{\mathrm{I}}(\mathbb{M},\mathbb{M}') + C_n(\delta)}, \tag{15}$$

*where $C_n(\cdot)$ is a continuous function such that $C_n(\delta) \to 0$ when $\delta \to 0$.*

*Proof.* Let $\mathbb{M} = \oplus_{i=1}^m M_i$ and $\mathbb{M}' = \oplus_{j=1}^{m'} M'_j$ be two persistence modules that are decomposable into intervals, let $x \in \mathbb{R}^k$ and let $0 \leq \delta \leq \min\{\delta(\mathbb{M}),\delta(\mathbb{M}')\}$.

**Notations.**    We first introduce some notations. Let $N$ (resp. $N'$) be the number of bars in $B_{\mathbf{e}_\Delta,x}$ (resp. $B'_{\mathbf{e}_\Delta,x}$), and assume without loss of generality that $N \leq N'$. Let $\Gamma$ be the partial matching achieving $d_{\mathrm{B}}(B_{\mathbf{e}_\Delta,x},B'_{\mathbf{e}_\Delta,x})$. Let $N_1$ (resp. $N_2$) be the number of bars in $B_{\mathbf{e}_\Delta,x}$ that are matched (resp. not matched) to a bar in $B'_{\mathbf{e}_\Delta,x}$ under $\Gamma$, so that $N = N_1 + N_2$. Finally, note that $B_{\mathbf{e}_\Delta,x} = \{\ell : \exists i \text{ such that } \ell \in \mathcal{C}(\mathrm{im}(\ell_{\mathbf{e}_\Delta,x}) \cap M_i) \text{ and } \mathrm{im}(\ell_{\mathbf{e}_\Delta,x}) \cap M_i \neq \emptyset\}$, where $\mathcal{C}$ stands for the set of connected components (and similarly for $B'_{\mathbf{e}_\Delta,x}$), and let $F_\Gamma : B_{\mathbf{e}_\Delta,x} \to B'_{\mathbf{e}_\Delta,x}$ be a function defined on all bars of $B_{\mathbf{e}_\Delta,x}$ that coincides with $\Gamma$ on the $N_1$ bars of $B_{\mathbf{e}_\Delta,x}$ that have an associated bar in $B'_{\mathbf{e}_\Delta,x}$, and that maps the $N_2$ remaining bars of $B_{\mathbf{e}_\Delta,x}$ to some arbitrary bars in the $(N' - N_1)$ remaining bars of $B'_{\mathbf{e}_\Delta,x}$.

**A reformulation of the problem with vectors.** We now derive vectors that allow to reformulate the problem in a useful way. Let $\hat{V}$ be the sorted vector of dimension $N$ containing all weights $\|\mathcal{N}(x,\sigma)\cdot f\|_1$, where $f$ is the interval function whose support is $T_\delta(\ell)\cap M_i$ for some $M_i$, where $\ell\in B_{\mathbf{e}_\Delta,x}$ is a connected component of $\mathrm{im}(\ell_{\mathbf{e}_\Delta,x})\cap M_i$. Now, let $\hat{V}'$ be the vector of dimension $N'$ obtained by concatenating the two following vectors:

- the vector $\hat{V}'_1$ of dimension $N$ whose $i$th coordinate is $\|\mathcal{N}(x,\sigma)\cdot f'\|_1$, where $f'$ is the interval function whose support is $T_\delta(\ell')\cap M'_j$ for some $M'_j$, and $\ell'\in B'_{\mathbf{e}_\Delta,x}$ is the image under $\Gamma$ of the bar $\ell\in B_{\mathbf{e}_\Delta,x}$ corresponding to the $i$th coordinate of $\hat{V}$, i.e., $\ell'=F_\Gamma(\ell)$ where $[\hat{V}]_i=\|\mathcal{N}(x,\sigma)\cdot f\|_1$ and $f$ is the interval function whose support is $T_\delta(\ell)\cap M_{i_0}$ for some $M_{i_0}$. In other words, $\hat{V}'_1$ is the (not necessarily sorted) vector of weights computed on the bars of $B'_{\mathbf{e}_\Delta,x}$ that are images (under the partial matching $\Gamma$ achieving the bottleneck distance) of the bars of $B_{\mathbf{e}_\Delta,x}$ that were used to generate the (sorted) vector $\hat{V}$.

- the vector $\hat{V}'_2$ of dimension $(N'-N)$ whose $j$th coordinate is $\|\mathcal{N}(x,\sigma)\cdot f'\|_1$, where $f'$ is an interval function whose support is $T_\delta(\ell')\cap M'_j$ for some $M'_j$, and $\ell'\in B'_{\mathbf{e}_\Delta,x}$ satisfies $\ell'\notin\mathrm{im}(F_\Gamma)$. In other words, $\hat{V}'_2$ is the vector of weights computed on the bars of $B'_{\mathbf{e}_\Delta,x}$ (in an arbitrary order) that are not images of bars of $B_{\mathbf{e}_\Delta,x}$ under $\Gamma$.

Finally, we let $V$ be the vector of dimension $N'$ obtained by filling $\hat{V}$ (whose dimension is $N\leq N'$) with null values until its dimension becomes $N'$, and we let $V'=\mathrm{sort}(\hat{V}')$ be the vector obtained after sorting the coordinates of $\hat{V}'$. Observe that:

$$|V_{\delta,\sigma}(\mathbb{M})(x)-V_{\delta,\sigma}(\mathbb{M}')(x)|=[V-V']_1=[V-\mathrm{sort}(\hat{V}')]_1 \tag{16}$$

**An upper bound.** We now upper bound $\left\|V-\hat{V}'\right\|_\infty$. Let $q\in[\![1,N']\!]$. Then, one has $[V]_q=\|\mathcal{N}(x,\sigma)\cdot f\|_1$, where $f$ is an interval function whose support is $T_\delta(\ell)\cap M_i$ for some $M_i$ with $\ell\in B_{\mathbf{e}_\Delta,x}$ if $q\leq N$ and $\ell=\emptyset$ otherwise; and similarly $[\hat{V}']_q=\|\mathcal{N}(x,\sigma)\cdot f'\|_1$, where $f'$ is an interval function whose support is $T_\delta(\ell')\cap M'_j$ for some $M'_j$ with $\ell'\in B'_{\mathbf{e}_\Delta,x}$. Thus, one has:

$$
\begin{aligned}
[V-\hat{V}']_q &= \big|\,\|\mathcal{N}(x,\sigma)\cdot f\|_1-\|\mathcal{N}(x,\sigma)\cdot f'\|_1\,\big| \\
&\leq \|\mathcal{N}(x,\sigma)\cdot f-\mathcal{N}(x,\sigma)\cdot f'\|_1 \text{ by the reverse triangle inequality} \\
&= \|\mathcal{N}(x,\sigma)\cdot(f-f')\|_1 \text{ by linearity} \\
&\leq \|\mathcal{N}(x,\sigma)\|_2\cdot\|f-f'\|_2 \text{ by Hölder's inequality} \\
&= (\sqrt{\pi}\sigma)^k\cdot\|f-f'\|_2
\end{aligned}
$$

Since $(f-f')$ is an interval function whose support is $(T_\delta(\ell)\cap M_i)\triangle(T_\delta(\ell')\cap M'_j)$, one has $\|f-f'\|_2=\sqrt{|(T_\delta(\ell)\cap M_i)\triangle(T_\delta(\ell')\cap M'_j)|}$. Given a segment $\ell$ and a vector $\mathbf{u}$, we let $\ell_{\mathbf{u}}$ denote the segment $\mathbf{u}+\ell$, and we let $\ell^{\mathbf{u}}$ denote the (infinite) line induced by $\ell_{\mathbf{u}}$. More precisely:

$$
\begin{aligned}
\|f-f'\|_2^2 &= |(T_\delta(\ell)\cap M_i)\triangle(T_\delta(\ell')\cap M'_j)| \\
&= |\bigcup_{\mathbf{u}}(\ell^{\mathbf{u}}\cap M_i)\triangle\bigcup_{\mathbf{u}}((\ell')^{\mathbf{u}}\cap M'_j)| \\
&\qquad \text{where } \mathbf{u} \text{ ranges over the vectors such that } \|\mathbf{u}\|_\infty\leq\delta,\langle\mathbf{u},\mathbf{e}_\Delta\rangle=0 \\
&\leq \int_{\mathbf{u}}|(\ell^{\mathbf{u}}\cap M_i)\triangle((\ell')^{\mathbf{u}}\cap M'_j)|\mathrm{d}\mathbf{u} \\
&\leq \int_{\mathbf{u}}|(\ell^{\mathbf{u}}\cap M_i)\triangle(\ell\cap M_i)_{\mathbf{u}}|+|(\ell\cap M_i)_{\mathbf{u}}\triangle(\ell'\cap M'_j)_{\mathbf{u}}|+|(\ell'\cap M'_j)_{\mathbf{u}}\triangle((\ell')^{\mathbf{u}}\cap M'_j)|\mathrm{d}\mathbf{u} \\
&\leq \int_{\mathbf{u}}4d_{\mathrm{B}}(B_{\mathbf{e}_\Delta,x},B_{\mathbf{e}_\Delta,x+\mathbf{u}})+4d_{\mathrm{B}}(B_{\mathbf{e}_\Delta,x},B'_{\mathbf{e}_\Delta,x})+4d_{\mathrm{B}}(B'_{\mathbf{e}_\Delta,x},B'_{\mathbf{e}_\Delta,x+\mathbf{u}})\mathrm{d}\mathbf{u} \tag{17} \\
&\leq 4\int_{\mathbf{u}}\|\mathbf{u}\|_\infty+d_{\mathrm{I}}(\mathbb{M},\mathbb{M}')+\|\mathbf{u}\|_\infty\,\mathrm{d}\mathbf{u} \text{ by Lemma 1 and 2}
\end{aligned}
$$

Inequality (17) comes from the fact that the symmetric difference between two bars (in two different barcodes) that are both matched (or unmatched) by the optimal partial matching is upper bounded by four times the bottleneck distance between the barcodes, and that (by assumption) the partial matchings achieving $d_{\mathrm{B}}(B_{\mathbf{e}_\Delta,x}, B_{\mathbf{e}_\Delta,x+\mathbf{u}})$ and $d_{\mathrm{B}}(B'_{\mathbf{e}_\Delta,x}, B'_{\mathbf{e}_\Delta,x+\mathbf{u}})$ are induced by $\mathbb{M}$ and $\mathbb{M}'$.

**Conclusion.** Finally, one has

$$
\begin{aligned}
|V_{\delta,\sigma}(\mathbb{M})(x) - V_{\delta,\sigma}(\mathbb{M}')(x)| &= [V - V']_1 = [V - \mathrm{sort}(\hat{V}')]_1 \text{ from Equation (16)}\\
&\leq \|V - V'\|_\infty \\
&\leq (\sqrt{\pi}\sigma)^k \cdot \sqrt{2^{n+1}\delta^{n-1} d_{\mathrm{I}}(\mathbb{M}, \mathbb{M}') + C_n(\delta)}, \quad (18)
\end{aligned}
$$

with $C_n(\delta) = 8 \int_{\mathbf{u}} \|\mathbf{u}\|_\infty \, \mathrm{d}\mathbf{u} \to 0$ when $\delta \to 0$. Inequality (18) comes from the fact that any upper bound for the norm of the difference between a given vector $\hat{V}'$ and a sorted vector $V$, is also an upper bound for the norm of the difference between the sorted version $V'$ of $\hat{V}'$ and the same vector $V$ (see Lemma 3.9 in [12]). $\qquad\square$

While the stability constant is not upper bounded by $\delta$, $V_{\delta,\sigma}$ is more difficult to compute than the S-CDRs presented in Definition 2.

## H  Pseudo-code for S-CDRs

In this section, we briefly present the pseudo-code that we use to compute S-CDRs. Our code is based on implicit descriptions of the candidate decompositions of multiparameter persistence modules (which are the inputs of S-CDRs) through their so-called *birth* and *death* corners. These corners can be obtained with, e.g., the public softwares `MMA` [29] and `Rivet` [28].

In order to compute our S-CDRs, we implement the following procedures:

1. Given an interval module $M$ and a rectangle $R \subset \mathbb{R}^n$, compute the restriction $M\big|_R$.

2. Given an interval module $M$ (or $M\big|_R$), compute $d_{\mathrm{I}}(M, 0)$. This allows to compute our weight function and first interval representation in Definition 2.

3. Given an interval module restricted to a rectangle, compute the volume of the biggest rectangle in the support of this module. This allows to compute the third interval representation in Definition 2.

For the first point, Algorithm 1 works by "pushing" the corners of the interval on the given rectangle in order to obtain the updated corners.

---

**Algorithm 1:** Restriction of an interval module to a rectangle

---

**Data:** birth and death corners of an interval module $M$, rectangle $R = \{z \in \mathbb{R}^n : m \le z \le M\}$

**Result:** new_interval_corners, the birth and death corners of $M\big|_R$.

**for** interval $= \{$interval_birth_corners, interval_death_corners$\}$ ***in*** $M$ **do**

    new_birth_list $\leftarrow [\,]$;

    **for** $b$ ***in*** interval_birth_corners **do**

        **if** $b \le M$ **then**

            $b' = \{\max(b_i, m_i)$ for $i \in [\![1, n]\!]\}$;

            Append $b'$ to new_birth_list;

        **end**

    **end**

    new_death_list $\leftarrow [\,]$;

    **for** $d$ ***in*** interval_death_corners **do**

        **if** $d \ge m$ **then**

            $d' = \{\min(d_i, M_i)$ for $i \in [\![1, n]\!]\}$;

            Append $d'$ to new_death_list;

        **end**

    **end**

    new_interval_corners $\leftarrow$ [new_birth_list, new_death_list];

**end**

---

For the second point, we proved in Lemma 3 that our S-CDR weight function is equal to $d_{\mathrm{I}}(M, 0)$ and has a closed-form formula with corners, that we implement in Algorithm 2.

---

**Algorithm 2:** S-CDR weight function

---

**Data:** birth and death corners of an interval module $M$

**Result:** distance, the interleaving distance $d_I(M, 0)$.

distance $\leftarrow 0$;

**for** $b$ ***in*** $M$_birth_corners **do**

    **for** $d$ ***in*** $M$_death_corners **do**

        distance $\leftarrow \max\left(\text{result}, \frac{1}{2}\min_i(d_i - b_i)_+\right)$;

    **end**

**end**

---

The third point also has a closed-form formula with corners, leading to Algorithm 3.

---

**Algorithm 3:** S-CDR interval representation

---

**Data:** birth and death corners of an interval module $M$

**Result:** volume, the volume of the biggest rectangle fitting in $\mathrm{supp}\,(M)$

volume $\leftarrow 0$;

**for** $b$ ***in*** $M$_birth_corners **do**

    **for** $d$ ***in*** $M$_death_corners **do**

        volume $\leftarrow \max\left(\text{result}, \Pi_i(d_i - b_i)_+\right)$;

    **end**

**end**

---

Finally, we show how to get the persistence barcodes corresponding to slices of an interval module solely from the corners of the interval module.

---

**Algorithm 4:** Restriction of an interval module to a line

---

**Data:** birth and death corners of an interval module $M$, a diagonal line $l$

**Result:** barcode, the persistence barcode associated to $M\big|_l$

barcode $\leftarrow [\,]$;

$y \leftarrow$ an arbitrary point in $l$;

**for** interval $= \{\text{interval\_birth\_corners}, \text{interval\_death\_corners}\}$ *in* $M$ **do**

    birth $\leftarrow y + \mathbf{1} \times \min_{b \in \text{interval\_birth\_corners}} \max_i b_i - y_i$;

    death $\leftarrow y + \mathbf{1} \times \max_{d \in \text{interval\_death\_corners}} \min_i d_i - y_i$;

    bar $\leftarrow [\text{birth}, \text{death}]$;

    Append bar to barcode;

**end**

---

## I  UCR acronyms

| Dataset | Acronym |
|---|---|
| `DistalPhalanxOutlineAgeGroup` | DPOAG |
| `DistalPhalanxOutlineCorrect` | DPOC |
| `ProximalPhalanxOutlineAgeGroup` | PPOAG |
| `ProximalPhalanxOutlineCorrect` | PPOC |
| `ProximalPhalanxTW` | PPTW |
| `ItalyPowerDemand` | IPD |
| `GunPoint` | GP |
| `GunPointAgeSpan` | GPAS |
| `GunPointMaleVersusFemale` | GPMVF |
| `PowerCons` | PC |

