# OpenReview forum: "A Framework for Fast and Stable Representations of Multiparameter Persistent Homology Decompositions"
_NeurIPS.cc/2023/Conference — NeurIPS 2023 poster_

### Official Review · Reviewer_HVag · 2023-06-24

**Soundness:** 4 excellent
**Presentation:** 4 excellent
**Contribution:** 3 good
**Rating:** 7
**Confidence:** 5

**Summary:**

Persistent homology (PH) is the most important method in topological data analysis (TDA), and multiparameter persistence (MPH) is its natural generalization which is expected to significantly boost its performance. However, because of several mathematical roadblocks, MPH could not be effectively used in real life applications. In this work, the authors proposes a general framework to vectorize MPH. Their framework generalizes all the known existing work as special cases. Furthermore, they provide a subset of vectorizations which are stable. Finally, another obstruction to use MPH in real life applications was their computational costs. The authors new framework provides a much faster way to obtain these vectorizations.

**Strengths:**

Framework provided is very general and it addresses a critical need in TDA.

Stable MPH vectorizations, and their convergence studies are excellent contribution to the theory of TDA.

Computationally feasible MPH vectorizations will finally enable to use MPH effectively in real life applications.

**Weaknesses:**

Experiments section could be more detailed, and more in-depth analysis could have been provided for the performance of these vectorizations in real life applications. In particular, the authors compare the performance of their vectorizations only with respect to other TDA models, and some simple other models in point cloud setting. Instead, more thorough analysis could have been provided to compare these MPH vectorizations with respect to SOTA models from different domains.

A performance evaluation in graph setting would be nice as it is another very important application area of PH.


**Questions:**

From theoretical standpoint, stable vectorizations are always preferable, however, from ML side, when enforcing stability, we might be losing too much information and scarifying performance. With this in mind, can you also suggest a practical, computationally efficient T-CDR methods which potentially provide better performance?

**Limitations:**

As authors stated, one of the main limitation of the approach is coming from the generality/flexibility of their framework. For ML applications, there are several choices to make (hyperparameter tuning) to define a suitable vectorization for a given situation.

---

> ### Author Rebuttal · Authors · 2023-08-10
>
>  *Experiments section could be more detailed, and more in-depth analysis could have been provided for the performance of these vectorizations in real life applications. In particular, the authors compare the performance of their vectorizations only with respect to other TDA models, and some simple other models in point cloud setting. Instead, more thorough analysis could have been provided to compare these MPH vectorizations with respect to SOTA models from different domains.*
>
> Thank you for this suggestion. Given that our goal was mostly to show improvement over the topological baselines, we put less emphasis on the non topological ones, as advocated also in the cited earlier works. However, we added a new experiment on graph data, for which MPH is known to work well, and that incorporates more SOTA baselines. See general comment and Table 3 in the rebuttal pdf.
>
> *A performance evaluation in graph setting would be nice as it is another very important application area of PH.*
>
> Thank you for this suggestion. We have added a new experiment on graph data, which shows that our framework can be applied to more general structured data than geometric point clouds. See our answer above and our general comment to all reviewers.
>
> *From theoretical standpoint, stable vectorizations are always preferable, however, from ML side, when enforcing stability, we might be losing too much information and scarifying performance. With this in mind, can you also suggest a practical, computationally efficient T-CDR methods which potentially provide better performance?*
>
> Thank you for this suggestion. Notice that the parameters we use in the experiments ($p=0,1$) are already not stable for the interleaving distance between persistence modules, but only for the bottleneck distance, as per Theorem 1. One future direction is indeed to investigate other types of T-CDR parameters: an interesting step in this path would be to borrow descriptors from the computer vision literature (such as, e.g., spin images from `Using Spin Images for Efficient Object Recognition in Cluttered 3D Scenes` (Johnson, Hebert, 1999)) in order to capture more information (even if it deteriorates stability) about the module summands of the decomposition in the $w$ or $\phi$ T-CDR parameters. We will make this clear in the text.

---

> > ### Comment · Reviewer_HVag · 2023-08-13
> >
> > Thank you for your answers and additional experiments. I have no further questions. Good luck with your submission.

---

### Official Review · Reviewer_BEmH · 2023-07-06

**Soundness:** 3 good
**Presentation:** 4 excellent
**Contribution:** 3 good
**Rating:** 6
**Confidence:** 3

**Summary:**

The paper investigated the persistent homology with multiple filtration parameters. The authors proposed a framework, T-CDR, which generalized the previous studies in multi-parameter persistent homology. They further presented stability and convergence guarantees on S-CDR, which is a special case of T-CDR introduced to ensure robustness. Third, their theoretical claim and contributions were supported by the empirical convergence studies as well as classification tasks on several immunohistochemistry datasets.


**Strengths:**

1. [Originality] The proposed framework, T-CDR in Definition 1 not only generalized previous work in candidate decomposition (e.g., MPI) as well as approaches in rank-invariant (MPL, MPK). I believe that it opens up new research on ensuring different guarantees by varying parameters/operators in (1).
1. [Significance] The stability and convergence guarantee (Theorem 1 and (8)) of S-CDR, which is a special case of T-CDR, provides a more stable and useful multi-parameter persistent homology framework.
1. [Quality] The authors supplement the convergence rate claim with informative empirical convergence studies, showing a clear trend of error rate reduces with $~n^{-1/2}$ as $n$ grows matching eq. (8).


**Weaknesses:**

1. In the empirical convergence rate experiment, it was not clear to me what is the “ground truth” representation you are comparing against. For synthetic data in Figure 3, it makes sense that you can get access to the density $f$ (therefore $\mathcal F_{C, f}$ and $\mathbb M$) given that you generated the data from some probability distribution. How do you get the $\mathbb M$ for the immunohistochemistry data (as in Figure 4)? Provide more clarifications on this will be beneficial.
1. It seems like MPL runtime can be improved 25x-50x with the sparse implementation in Algorithm 4 (see, e.g., Row #2 of Table 2 vs. Rows #4 and #6 of Table 2); this suggests that the runtime win might be due implementation rather than faster algorithmic time complexity. The claim will be more convincing if the authors can provide more insights, justifications, or analyses of the time complexity as to why the proposed algorithm is more efficient than prior work.
1. What is the bifiltration parameters used for each experiments in Sections 4? Are they radius and co-density for Figure 3, and CD8 and CD68 for the immunohistochemistry data? Adding more explanation there will increase the clarity more.
1. In Sections 1-3, the function $f$ is used to define the multi-parameter filtration function; however, in Section 4, the notation is defined as the density (e.g., in L254). I would suggest to choose another notation to avoid confusion.


**Questions:**

1. It looks to me most of Section 4.1 is a continuation of Theorem 1, is there any specific reason to put this in this section rather than in Section 3 (and make it a Proposition/Corollary)?
1. Empirically, how sensitive is the algorithm for the larger intrinsic dimension $d$? If I understand the experiments correctly, they all seem to have $d=2.$ I am curious about how the intrinsic dimension $d$ will impact convergence and/or performance.
1. I am also curious about how the proposed framework can be applied in higher-order homology descriptors (empirically). This is also related to Question #2 above.
1. TDA has been applied in numerous different domains such as galaxy, proteins, single-cell sequencing, 3D CAD point clouds, medical imaging [A-C] etc. I am curious if the proposed multi-parameters filtraion work can be expanded in fields outside immunohistochemistry.
1. How optimal/tight is the bound in (8)? Can we get a better convergence result if we choose op, $\omega$, and/or $\phi$ differently?
1. [Minor] Given that T-CDR is a generalization of both the rank-invraint and candidate decomposition method, should the name template “candidate decomposition” representation be modified to better reflect what it can be capable of?
1. [Minor/Typo?] Should the citation in L477 be [CB20] instead of the PersLay paper?

---
[A] Wasserman, Larry. “Topological Data Analysis.” Annual Review of Statistics and Its Application 5 (2018): 501–32.

[B] Chen, Yu-Chia, and Marina Meila. “The Decomposition of the Higher-Order Homology Embedding Constructed from the k-Laplacian.” Advances in Neural Information Processing Systems 34 (2021).

[C] Wu, Pengxiang, Chao Chen, Yusu Wang, Shaoting Zhang, Changhe Yuan, Zhen Qian, Dimitris Metaxas, and Leon Axel. “Optimal Topological Cycles and Their Application in Cardiac Trabeculae Restoration.” In International Conference on Information Processing in Medical Imaging, 80–92. Springer, 2017.


**Limitations:**

Authors have addressed the limitations of their work. Negative social impact is not applicable because this work is a theoretical contribution.

---

> ### Author Rebuttal · Authors · 2023-08-10
>
>  *In the empirical convergence rate experiment, [...] How do you get the for the immunohistochemistry data (as in Figure 4)?*
>
> This is a very good question. As you mentioned, we do not know the true distribution, so the target that we use (and that we measure the distance to) is the S-CDR obtained from the density computed with a KDE that was fit on the whole point cloud. We then generated the curves by drawing several subsamples, and measuring the differences between the S-CDRs of the KDE fit on the subsamples and the target S-CDR. Note that subsampling in this way is heuristically justified by the theory of the bootstrap/ resampling methods.
>
>  *the authors can provide more insights, justifications, or analyses of the time complexity as to why the proposed algorithm is more efficient than prior work.*
>
> This is a very good observation: indeed, concerning MPL, the huge improvement we observe is definitely due to the data structure we use; while the code in `multipers` needs all barcodes be stored in a heavy list of arrays, we use an implicit, sparse representation of candidate decompositions that allows for fast queries of barcodes and thus efficient MPL and MPI computations. We believe this is also what makes our implementation of T-CDRs and S-CDRs quite fast, in addition to their already favorable theoretical complexity (which is roughly equal to the product of the numbers of birth and death corners of our implicit representations of modules). We will make this more clear in the text.
>
> *What is the bifiltration parameters used for each experiments in Sections 4?*
>
> We always used Rips + codensity (estimated with kernel density estimators (KDE)), which is more stable yet very similar to the filtrations used in earlier works. Concerning the immunohistochemistry data, CD8 and CD68 were used only to create several different point clouds with their own labels (e.g., the point cloud of CD8 cells, with label "CD8", and the one of CD68 cells, with label "CD68"), and then they were all processed with Rips + codensity filtrations. We will make this clear in the text. See also our answer to question 3 of reviewer MKJT.
>
> *is there any specific reason to put this in this section rather than in Section 3 [...]?*
>
> Thank you for this suggestion. We initially wanted to emphasize the fact that S-CDRs and T-CDRs are general representations that can handle any candidate decompositions, while results in Section 4.1 are specific to the Rips + codensity filtrations.
>
> *Empirically, how sensitive is the algorithm for the larger intrinsic dimension $d$ ?*
>
> This is a very good question. In terms of convergence, we did some new experiments on point clouds in three dimensions, see Figure 1 in the rebuttal pdf, and did observe convergence with specific rates. In terms of classification performance, notice that while immunohistochemistry data is in two dimensions, UCR data was processed with time delay embedding with a length $3$ window, which produced point clouds in dimension $3$. In terms of running time performances, the intrinsic dimension only has an impact on the candidate decomposition methods (e.g., `MMA`, `Rivet`, etc) and/or the filtration values (e.g., density computed with KDE), but no impact on our T-CDRs and S-CDRs, as they only process persistence modules and are oblivious to how these modules were computed.
>
> *I am also curious about how the proposed framework can be applied in higher-order homology descriptors (empirically).*
>
> We are a bit unsure of what is meant by higher-order homology.
> --- If this refers to the homology dimension (e.g., $H_1$ vs. $H_{2}$), note that S-CDRs and T-CDRs can be applied straightforwardly as they are oblivious to the homology degrees and only needs persistence modules. We also run new convergence experiments to check whether convergence was still occurring in $H_2$ for a synthetic data set in $R^3$ in Figure 1 of the rebuttal pdf; while rates can be different than homology dimensions $0$ and $1$, we did always observe convergence happening.
> --- If this refers to the number of filtrations, the only limitation is the computational complexity, as the definitions of the T-CDRs and S-CDRs apply in any dimension.
>
>  *I am curious if the proposed multi-parameters filtraion work can be expanded in fields outside immunohistochemistry.*
>
> We have added a new experiment on graph data, which shows that our framework can be applied outside of geometric point clouds. See Table 3 in the rebuttal pdf and our general comment to all reviewers.
>
> *How optimal/tight is the bound in (8)? Can we get a better convergence result [...]?*
>
> We believe our bound is not necessarily optimal. Roughly speaking, our bound is obtained with:
>        - Stability of interleaving between Prokhorov and Wasserstein (a)
>        - Control of the Haussdorff distance (b) and of density estimation (c)
> Note that (a) is tight by the universality property (see section 5.2 in `The Theory of the Interleaving Distance on Multidimensional Persistence Modules` (Lesnick, 2015)). However, it should be possible to improve the bound by finding a specific class of data sets that would provide a tighter control of either (b) or (c). Whether it is possible to also get a better bound by fine tuning T-CDR parameters is a very interesting suggestion that we plan to study in future work.
>
> *Should the name template “candidate decomposition” representation be modified to better reflect what it can be capable of?*
>
> Thanks for the suggestion. Since T-CDR can only work once a decomposition has been provided (wherever it comes from), we believe that it is important that this appears in the representation name.
>
> *Should the citation in L477 be [CB20] instead of the PersLay paper?*
>
> Are you referring to L177 (the paper only has 458 lines)? In this case, we actually meant PersLay, since $w'$ and $\phi'$ are applied on restrictions of the persistence module to lines, or, in other words, to single-parameter persistence diagrams.

---

> > ### Comment · Reviewer_BEmH · 2023-08-17
> >
> > Thank you for the detailed response!
> >
> > - For "higher-order homology descriptors", yes, I am referring to homology dimension. Thanks for the additional experiments, I have no further question in this.
> > - I am referring to L477 in the appendix. Quote what you write there as following: "In this section, we provide theoretical and experimental evidence that the multiparameter persistence image (MPI) [CCI+20], which is another decomposition-based...". Given that you are talking about MPI, I naturally think that you should cite [CB20] rather than [CCI+20]. Maybe this is a typo?
> >
> > The rest look good!

---

> > > ### Author Response · Authors · 2023-08-17
> > > **Answer to reviewer BEmH**
> > >
> > > Yes you are totally right, thank you for noticing this mistake! We will put the correct reference [CB20] instead.
> > >
> > > Thanks again for all your comments and suggestions on our work.

---

### Official Review · Reviewer_MKJT · 2023-07-06

**Soundness:** 3 good
**Presentation:** 4 excellent
**Contribution:** 4 excellent
**Rating:** 7
**Confidence:** 3

**Summary:**

As the title clearly suggests, a general representation of multiparameter persistent homology (MPH) is introduced. Unlike earlier representations that either yield a loss in information or are unstable, the proposed vectorization is strictly more informative, and is shown to be stable for a specific settings of the parameters in the general approach. The approach is evaluated on several real data sets.

**Strengths:**

(S1) I do not have an overview of MPH literature, but assuming that the authors provide a complete and honest overview of earlier work, the contributions in this paper are substantial.

(S2) Figure 1 and the discussion on Lines 170 – 185 clearly and nicely position the work in the existing literature.


**Weaknesses:**

The main experimental results in Table 1 show that the proposed S-CDR approach is half of the time significantly outperformed by other existing methods.

**Questions:**

(Q1) Looking at Figure 2, it does not seem as challenging, but rather straightforward to represent the plot in the middle with the image on the right?

(Q2) You claim that S-CDR is strictly more powerful than MPK, but MPK achieves better scores on DPOC, PPTW, GPAS, GPMVF data sets (Table 1). This deserves at least a brief discussion.

(Q3) Why do you exclude some data sets considered in [CB20] from your experiments (Table 1)? Are the filtrations used to calculate your S-CDR the same as the filtrations to calculate MPK, MPL and MPI in earlier works? Why do you not evaluate the running times also on the Table 1 data set, and compare it against the results in [CB20]?



Other minor comments:

-	The order of Figures 1 and 2 should be reversed, since the latter is referenced first in the text? Similarly, Sections 4.2 and 4.3, or Table 1 and Table 2 should be reversed?
-	When listing the contributions, add “(Section 3)” after “statistical convergence of our representations”. The Outline paragraph soon below then becomes almost redundant.
-	Line 169, Line 177, Line 343: Perslay -> PersLay
-	Line 202: “the two following S-CDR”. I would remove “two,” as it might imply a diminished contribution, since you actually define an S-CDR for every p in N.
-	I would start every item in a list on Lines 194-195 and Line 204 on a new line, as this is important information that the reader should be able to find and read easily. If there are space limitations, you can sacrifice the Outline paragraph in Section 1, see one of the minor comments above.
-	Table 1: What is highlighted in red?
-	Line 314: “the same bifiltration as in the previous section”. Subsection? In this case too, the previous subsection does not contain any information on bifiltration.
-	Table 1: What does “P” denote for the last row in the table?
-	Line 326: “almost always outperform” is an overstatement.
-	Table 2: For better readability, only consider or s or ms, remove them from the individual cells and place in brackets in column titles.
-	Line 344: parameter -> parameters
-	References: Be consistent between capital case vs. lower case for the paper titles. Remove # from 2D, and remove double reference [CdGO15], [CdSGO15].


**Limitations:**

The limitations and future research directions are clearly presented in the final section.

---

> ### Author Rebuttal · Authors · 2023-08-10
>
> *The main experimental results in Table 1 show that the proposed S-CDR approach is half of the time significantly outperformed by other existing methods.*
>
> This is correct.  The table indicates that for some time series and graph tasks, topological methods are less effective than "classical" methods; in these cases, the close scores of the topological methods indicate limits of what TDA can achieve.  The real strength of the topological methods (and notably S-CDR) can be seen from the results on the immunohistochemistry data.
>
> *Looking at Figure 2, it does not seem as challenging, but rather straightforward to represent the plot in the middle with the image on the right?*
>
> Thanks for the suggestion. We will make the required changes.
>
>  *You claim that S-CDR is strictly more powerful than MPK, but MPK achieves better scores on DPOC, PPTW, GPAS, GPMVF data sets (Table 1). This deserves at least a brief discussion.*
>
> You are right. We believe that S-CDR shines particularly when candidate decompositions of modules matter the most (w.r.t. rank-invariant based methods). We believe this is what happens for the immunohistochemistry data (and what explains the improvement), while UCR data can be classified using only the information encoded in the rank invariant, so that the extra content in the candidate decompositions and our corresponding S-CDRs is not used so much, leading to comparable yet not strongly superior accuracies. We will update the text to make this clear.
>
> *Why do you exclude some data sets considered in [CB20] from your experiments (Table 1)?*
>
> This is a very good question. Unfortunately, the immunofluorescence data presented in [CB20] is protected and has not been released publicly, so we could not reuse it. Instead, we provided experiments on a similar public data set from immunohistochemistry.
>
> *Are the filtrations used to calculate your S-CDR the same as the filtrations to calculate MPK, MPL and MPI in earlier works?*
>
> The filtrations we used are slightly different than the ones of earlier work for MPK, MPL and MPI, but they are very similar: earlier work use Alpha + Distance-to-Measure (DTM) filtrations while we use Rips + codensity (computed with kernel density estimators (KDE)) filtrations. For sake of completeness and fairness, we rerun the experiments using also Alpha and DTM filtrations, see Table 1 in the rebuttal pdf; we did not observe drastic changes in the corresponding results, and the main takeaway of the table stays the same: S-CDRs are quite competitive with both topological and non topological baselines.
>
> The reason we initially used different filtrations is the following: despite being theoretically related, Alpha and Rips filtrations differ in the final complex they filter. Indeed, on a dataset of n points, Rips filters the complete, abstract simplex with n vertices, while Alpha filters (roughly speaking) the Delaunay triangulation of the points. This induces some lack of stability and convergence rates for Alpha + DTM in the multi-parameter setting, as some critical simplices might never appear in the single-parameter filtrations of some lines in the module if their densities are too small, leading to infinite bars, while the complete simplex associated to Rips contains so many faces that every cycle is eventually closed by some simplex at some point.
>
> *Why do you not evaluate the running times also on the Table 1 data set, and compare it against the results in [CB20]?*
>
> This is a very good question. The running times improvements for UCR data sets (in Table 1) were not particularly striking due to the quite small data set sizes (leading to running times that are quite small for all competitors) so we did not include them. For sake of completeness, we put them back (for a few representative UCR data sets) in Table 2 of the rebuttal pdf.
>
> *What is highlighted in red?*
>
> Red and bold numbers correspond to the best scores in each category (non-topological baselines / single-parameter persistence / multi-parameter persistence) for the UCR data sets, and to the best score overall for the immunohistochemistry data. We will make this clear in the text. See also our answer to reviewer pDMj.
>
> *What does “P” denote for the last row in the table?*
>
> "P" stands for single-parameter persistence. Since all three such methods ("PSS-K", "P-I", "P-L") provided the (almost) same result, we only provided one number in order to save space.
>
> Thank you very much also for all the minor suggestions and comments concerning the text. We will make the required changes.

---

> > ### Comment · Reviewer_MKJT · 2023-08-14
> >
> > Thank you for the detailed response!

---

### Official Review · Reviewer_pDMj · 2023-07-10

**Soundness:** 3 good
**Presentation:** 3 good
**Contribution:** 3 good
**Rating:** 6
**Confidence:** 2

**Summary:**

The paper addresses topological data analysis in the multiparameter setting and focuses on the problem of representation of multiparameter persistent homology by vector space elements so that standard ML algorithms can be applied.  The key advance in the paper is the leveraging of decomposition ideas in multiparameter settings to devise a representation framework that is shown to be theoretically stable with practical, efficient computation.  The  strengths in statistical convergence, accuracy and computational speed are validated in specific UCR classification datasets.


**Strengths:**

Originality:
The paper draws inspiration from past work on multiparameter persistent homology by Carriere et al. It offers a theoretical formulation that provides stability guarantees (e.g. statistical rates of convergence) and is a generalization of the past work.

Quality:  The paper is comprehensive including the appendix.  I did not go through the details of the proofs and cannot comment on the theoretical correctness of the paper.

Clarity:  The paper is very clear with comprehensive references to the latest in the field.

Significance:  The paper appears to be of theoretical significance and will be of importance to researchers in Topological Data analysis.

**Weaknesses:**

While the experimental results show the power of S-CDR on several datasets,  It will be interesting to see how the method applied to more complex datasets (e.g. from imaging related applications). The experimental validation is limited but sufficient.

**Questions:**

I could not follow the notation in table 1 as it is not clear why there are multiple highlighted numbers (I would assume that the red numbers correspond to the best classification results, but in both table 1 and in the appendix on time series classifications there are multiple flagged columns of relevance).

---

> ### Author Rebuttal · Authors · 2023-08-10
>
> *Experiments on more complex datasets, e.g., imaging related applications. The experimental validation is limited but sufficient*
>
> We have included a second set of experiments on graph data that demonstrates that our representations can be successfully applied to other types of data sets. Results are available in Table 3 of the rebuttal pdf. See also our general comment to all reviewers.
>
> *Table 1 notations : clarify the highlighted numbers*
>
> We agree that our notations were confusing. Red and bold numbers correspond to the best scores in each of the three categories (non-topological baselines / single-parameter persistence / multi-parameter persistence) for the UCR data sets, and to the best score overall for the immunohistochemistry data. We will make more clear in the text by putting the best score overall in bold font and decorate (e.g., by underlining) the best in each category.

---

> > ### Comment · Reviewer_pDMj · 2023-08-16
> > **Satisfied with the rebuttal.**
> >
> > Thanks for the additional experiments and the revisions.

---

### Author Rebuttal · Authors · 2023-08-10

We first want to thank all reviewers for their careful and insightful reviews. We have responded to their comments, and will be happy to provide further explanations if needed.

We have also prepared a rebuttal pdf that contains a few more experiments that were rightfully suggested by the reviewers. If the paper is accepted, these new results will be included in the submission, either in the main paper or in the supplementary. In short,
1. we re-did classification experiments on UCR data sets and made sure we used the same filtrations as topological baseline competitors (Table 1),
2. we added the running times of our method for UCR data sets (Table 2),
3. we added a new classification experiments on graph data sets (Table 3),
4. we measured convergence rates for a point cloud in $R^3$ in homology degrees $0$ ,$1$ and $2$ (Figure 1)

More details about the new graph classification experiment: the results we provide were obtained after cross validating across several pairs and triplets of filtrations (including HKS, Ricci curvature and node degree), with the remaining parameters being the same than the ones we used for the UCR classification experiment. We compare S-CDRs to the Euler characteristic based multiparameter persistence methods ECP, RT, and HTn, introduced in `Euler characteristic tools for topological data analysis` (Hacquard, Lebovici, 2023). In order to also include non topological baselines, we also compare against the state-of-the-art graph classification methods RetGK introduced in `Retgk: Graph kernels based on return probabilities of random walks` (Zhang et al, NeurIPS 2018), FGSD introduced in `Hunt for the unique, stable, sparse and fast feature learning on graphs` (Verma, Zhang, NeurIPS 2017), and GIN introduced in `How powerful are graph neural networks?` (Xu et al, ICLR 2019).  These SOTA methods performed the best in the analysis of (Hacquard, Lebovici, 2023) so we decided to go with them and use the accuracies reported there. All methods are averaged over 10 folds with 90/10 training/test splits.

Overall, one can see from Table 3 in the rebuttal pdf that results are competitive with both topological and non-topological baselines; S-CDRs perform even slightly better on COX2. We believe this is yet another indication that our method can be applied successfully in quite general settings involving multiparameter topology.

---

### Decision · Program_Chairs · 2023-09-21

**Decision:**

Accept (poster)

**Comment:**

All reviewers agreed on the relevance and impact of the proposed work. As an expert in the topic myself, I am also **very excited** about the prospects of new methods that make multiparameter persistent homology available for a wider audience and a larger set of applications. The authors provided a comprehensive rebuttal and addressed some minor issues. I am thus absolutely confident that the paper is ready for publication and trust the authors to integrate the new results discussed during the rebuttal. It is my great pleasure to **endorse this paper for publication**.